# A Review on the Promising Plasma-Assisted Preparation of Electrocatalysts

**DOI:** 10.3390/nano9101436

**Published:** 2019-10-10

**Authors:** Feng Yu, Mincong Liu, Cunhua Ma, Lanbo Di, Bin Dai, Lili Zhang

**Affiliations:** 1School of Chemistry and Chemical Engineering, Shihezi University, Shihezi 832003, Chinaqdulmc@163.com (M.L.); mchua@shzu.edu.cn (C.M.); 2College of Physical Science and Technology, Dalian University, Dalian 116622, China; dilanbo@163.com; 3Institute of Chemical and Engineering Sciences, Agency for Science, Technology and Research, Jurong Island 627833, Singapore

**Keywords:** electrocatalyst, plasma, defect rich, surface etching, heteroatom doping

## Abstract

Electrocatalysts are becoming increasingly important for both energy conversion and environmental catalysis. Plasma technology can realize surface etching and heteroatom doping, and generate highly dispersed components and redox species to increase the exposure of the active edge sites so as to improve the surface utilization and catalytic activity. This review summarizes the recent plasma-assisted preparation methods of noble metal catalysts, non-noble metal catalysts, non-metal catalysts, and other electrochemical catalysts, with emphasis on the characteristics of plasma-assisted methods. The influence of the morphology, structure, defect, dopant, and other factors on the catalytic performance of electrocatalysts is discussed.

## 1. Introduction

Tremendous research efforts have been put to address global energy and environmental concerns. One of the promising energy solutions is electrocatalytic technology, such as fuel cells, the hydrogen production reaction, CO_2_ recycling, and ammonia synthesis (Figure 1) [1,2].

Electrocatalytic technology mainly involves two categories: (a) the electrocatalytic oxidation reaction; i.e., the oxygen evolution reaction (OER), hydrogenation reaction (HOR), and methanol oxidation reaction (MOR); and (b) the electrocatalytic reduction reaction; i.e., the oxygen reduction reaction (ORR), CO_2_ reduction reaction (CO_2_RR), N_2_ reduction reaction (NRR), hydrogen evolution reaction (HER), etc. Electrocatalytic reactions can be carried out at atmospheric temperature and pressure. For example, sustainable ammonia synthesis is achieved under carbon-free conditions using N_2_ and H_2_O as the raw materials; the electrochemical reduction of CO_2_ is used to prepare chemicals with high energy densities, such as C1 and Cn, electrochemically converting electrical energy into an efficient and clean energy carrier-hydrogen energy [3,4]. In addition, chemical energy can be converted into electrical energy using HOR, MOR, or ORR in a fuel cell. ORR and OER can also be used as reversible half-reactions for rechargeable metal-oxygen batteries, using metals, such as Li, Mg, and Zn as energy carriers rather than hydrogen [5].

Currently, noble metal catalysts (such as Au, Pt, Ru, and Ag) are the most important materials for electrocatalysts. They are characterized by easy adsorption of reactants on the surface, the ability to form active intermediates, excellent acid-base corrosion resistance, and outstanding catalytic activity, selectivity, and stability. Compared with noble metal electrocatalysts, inexpensive non-noble metal electrocatalysts also have excellent electrocatalytic performance, especially for transition metal alloys and compounds (oxides, sulfides, nitrides, phosphides, and carbides) [6]. Transition metals can bond with the reaction molecules to form a transition state with lower energy barrier, thus reducing the activation energy of the whole reaction path and accelerating the chemical reaction [7]. Considering that there are many kinds of non-noble metal electrocatalysts, electrocatalysts with special chemical compositions and physical structures can also be designed and prepared to meet the needs of different electrocatalytic reactions [8]. The excellent acid-base corrosion resistance of non-metallic electrocatalysts have attracted extensive attention of researchers, especially for carbon-based electrocatalysts. Conventional carbon-based molecules have no hollow orbitals, which makes it difficult to participate in the electrocatalytic reaction. However, the electrocatalytic performance of carbon-based materials can be effectively improved by introducing other heteroatoms or functional groups on the surface of carbon materials [9].

In recent years, the plasma-assisted preparation method has attracted more and more research interest and has been widely used in the synthesis and modification of electrocatalyst materials. Plasma is partially ionized gas consisting of electrons, ions, molecules, free radicals, photons, and excited species, all of which are active species for the preparation and treatment of catalysts [10]. Differently from the traditional preparation methods, plasma can generate redox species, surface etching, and element doping during the nucleation of catalysts and crystal growth, so as to prepare highly dispersed, small particle size, defect-rich electrocatalysts [11]. This paper reviews the recent progress in plasma-assisted preparation methods, and discusses the effect of plasma on the enhanced catalytic activity through improved dispersion of active species and active sites. Finally, challenges and a future perspective on the development of plasma-assisted preparation of electrocatalysts are provided.

## 2. Noble Metal Electrocatalysts

### 2.1. Plasma Enhanced Deposition

In 1991, Keijser et al. [12], Dutch scientists, first proposed plasma enhanced atomic layer deposition (PEALD) technology. As shown in Figure 2a–c, the common PEALD technologies include direct plasma-enhanced atomic layer deposition, remote plasma-enhanced atomic layer deposition and radical-enhanced atomic layer deposition [13,14]. In 2015, Ting et al. [15] used PEALD technology to deposit Pt nanoparticles with different particle sizes on the TiO_2_ surface layer of Ti thin films, and prepared Pt/TiO_2_ catalysts. It was found that the average particle size of Pt nanoparticles increased from 3 nm to 7 μm with the increase of deposition times (Figure 2d–i). Pt nanoparticles with a size of ~5 nm exhibited excellent catalytic activity for MOR with the best CO tolerance and electrochemical stability (Table 1), which had potential applications in methanol fuel cells.

In the same year, Yoshiaki et al. [16] produced highly ionized metal plasma by coaxial pulse arc plasma deposition (CAPD), which deposited Pt nanoparticles with average particle diameter of 2.5 nm on the carbon carrier (Ketjenblack carbon). Compared with the commercial 20% Pt/C, the catalytic activity and stability of the catalysts were significantly improved for MOR. Moreover, the catalysts exhibited excellent initial potential and half-wave potential for ORR performance. The half-wave potential was 0.87 V, which was higher than that of 5% Pt/C (0.78 V) and 20% Pt/C catalysts (0.84 V).

Plasma sputtering (IPS) is also used to prepare noble metal electrocatalysts. Grigoriev et al. [17] used Cabot carbon black (Vulcan XC-72), carbon nanotubes and nanofibers as substrates to deposit Pt and PtPd nanoparticles, which had potential electrochemical activity in fuel cells, water electrolysis cells and dual-function fuel cells. Falch et al. [18] deposited then Pt_x_Pd_y_ film on SiO_2_ substrates by magnetron enhanced plasma sputtering. It was found that Pt3Pd2 and PtPd4 exhibited good catalytic activity in the electrocatalytic oxidation of SO_2_. Compared with pure platinum (0.598 + 0.011 V, standard hydrogen electrode (SHE)), those two thin film materials had lower initial potential (0.587 + 0.004 V, SHE), which showed their potential application in SO_2_ oxidation.

### 2.2. Gas Plasma-Assisted Preparation

Plasma is a high energy state gas with electrons, ions, and free radicals. Common gases can be used to form plasma, such as H_2_, O_2_, CO_2_, Ar, NH_3_, and N_2_. When high energy plasma gas gets into contact with the surface of the material, it will lead to the physical and chemical changes of the material surface, such as chemical reduction, surface etching, the generation of surface active groups, etc. [19] In 2018, Ma et al. [20] irradiated Pt-based catalysts with high-energy electrons and activated ions in H_2_ plasma, which could simultaneously reduce Pt ions and graphene oxide (GO), as shown in Figure 3a. By adjusting the mass ratio of GO to multi-walled carbon nanotubes (MWCNT), Pt nanoparticles with different particle size distribution were obtained (Figure 3b–e). Pt/GNT had excellent MOR performance and anti-poisoning performance. The current density of Pt/GNT was 97.9 mA/mg, which was 2.2 times higher than that of commercial Pt/C (44.1 mA/mg). The current density of Pt/GNT was 691.1 mA/mg, which was much higher than that of commercial Pt/C (368.2 mA/mg).

By contrast, Xu et al. [21] found that Pt nanoparticles in Pt/CNTs-HP prepared by H_2_ plasma reduction of Pt-based precursor showed more uniform distribution on CNT with the particle size of about 2 nm. They had better MOR electrocatalytic activity than a traditional hydrogen reduced catalyst (Pt/CNTs-H) and NaBH_4_ reduced catalyst (Pt/CNTs-N). Precursors of noble metal catalysts can also be reduced by Ar plasma. Sui et al. [22] obtained Pt-Ir/TiC bimetallic electrocatalysts by Ar plasma reduction method, which had finer metal crystals and higher metal dispersion. The bimetallic electrocatalyst showed higher ORR/OER catalytic activity than that of the Pt-Ir/TiC electrocatalysts prepared by chemical reduction method.

Gas plasma can also be used to treat noble metal catalysts directly. Ipshita et al. [23] treated the Au@Pt catalyst with Pt loading of 1.75 ug/cm^2^ (corresponding to two layers of Pt atoms covering a gold core with a diameter of 5 nm) by Ar plasma. It was found that Ar plasma can enhance the formation of highly active Pt (110) crystal face. The total surface area was as high as 48 ± 3 m^2^/g, which was equivalent to 44% of the utilization rate of Pt atoms. The catalysts exhibited excellent CO poisoning resistance and outstanding MOR and ORR properties. Koh et al. [24] prepared “Au Islands” catalysts by treating gold foil with O_2_ plasma, which showed excellent CO_2_RR performance. Compared with polycrystalline gold electrode, the “Au Islands” catalysts had excellent CO selectivity, and their Faraday efficiency was more than 95%, which attracted people’s attention.

### 2.3. Solution Plasma Sputtering

Solution plasma sputtering (SPS) can occur between electrodes and solutions, providing a new plasma-liquid interface that initiates a variety of physical and chemical processes, as shown in Figure 4a,b. The unique interaction can be used to prepare noble metal nanoparticles and to promote carbon aggregation, so as to produce excellent electrocatalysts [25,26]. Kim et al. [27] obtained metal filtrate through corrosion of electrodes by discharging in water with pulsed plasma. Pt and Pt-M (M = Cu, Ag and Pd) bimetallic nanoparticles were prepared, which had no element segregation or phase segregation during the alloying of Pt and M. Among them, Pt-Ag bimetallic nanoparticles exhibited excellent MOR electrocatalytic activity, stability, and durability. At the same time, Kim et al. [28] pointed out that the corrosion of anode electrode was more aggressive than that of cathode electrode during plasma discharge, and the composition of Pt-Pd bimetallic nanoparticles can be varied with power and electrode structures. Cho et al. [29] found that Pt_69_Pd_31_ had excellent catalytic activity for MOR, with a current density of 6.81 mA/cm^2^, making it an ideal candidate catalyst for methanol fuel cell.

Metal-carbon composites can be obtained by adding carbon materials to the metal filtrate prepared by SPS. In 2017, Zhang et al. [30] used SPS technology to synthesize PtPd filtrate directly from Pt and Pd wire, and then Koqin black carbon material was added to obtain PtPd/C. As shown in Figure 4c–f, PtPd/KB-2 prepared in a mixture of methanol and water had better dispersion of metal particles than that prepared in water. The particle diameter was about 2–5 nm. The MOR activity of PtPd/KB-2 (12 wt% Pt) was four times better than that of commercial Pt/C catalyst, and the electrochemical surface area was 2.5 times of that of commercial Pt/C catalyst. Meanwhile, the mass activity of PtPd/KB-2 after 300 cycles can reach 43%. In 2018, Horiguchi et al. [31] designed a SPS mobile cell, which can continuously add Vulcan XC72R to the solution containing Pt to prepare Pt/XC72 electrocatalysts, providing a new route for the continuous production of electrocatalysts.

Metal-carbon composites can also be obtained by placing metal nanoparticles prepared by SPS in uniformly dispersed carbon-containing suspension. In 2017, Huang et al. [32] acquired Pt/CoPt/MWCNTs composite catalysts by dispersing Pt/CoPt composite nanoparticles by SPS in a uniform aqueous solution of multi-walled carbon nanotubes (MWCNT). The electrocatalysts exhibited excellent MOR catalytic activity and good stability, which exhibited an activity of 1719 mA/mg_Pt_ 3.16 times higher than that of commercial Pt/C. It can be used as a catalyst for direct methanol fuel cell. Su et al. [33] prepared Pt nanoparticles with particle size of 2 nm by SPS in a XC72 suspension. Pt/C/TiO_2_ electrocatalysts were then synthesized by mixing the obtained Pt/C composites with TiO_2_ nanotubes under ultrasound. The catalysts showed good electrocatalytic activity, and the CO toxicity resistance of the current density was 315.2 mA/mg, which is 1.73 times higher than that of commercial Pt/C.

Besides, the carbon-containing suspension can be used instead of water or alcohol solvents to prepare carbon-supported precious metal electrocatalysts. Hu et al. [34] successfully prepared PdAu/KB electrocatalysts by plasma discharge in the suspension of Keqin black carbon material using Pd and Au wires as electrodes. PdAu alloy nanoparticles with an average particle size of 2–5 nm were uniformly distributed on KB, and exhibited outstanding ORR activity in an acidic solution (0.5 M H_2_SO_4_, 240 cycles) and an alkaline solution (0.5 M NaOH, 700 cycles). As electrocatalysts, PdAu alloy nanoparticles have potential applications in future fuel cells or metal air batteries.

In addition to water or alcohol solvents, carbon-supported noble metal electrocatalysts can be directly obtained by using organic solutions as liquid plasma discharge media. As shown in Figure 5a–b, carbon nanospheres (CNS) can be formed when benzene is used as an organic solvent in plasma discharge. At the same time, metal nanoparticles (such as Au and Pt) produced by metal electrode sputtering are deposited on carbon nanospheres to obtain highly active electrocatalysts, loaded with precious metal nanoparticles. At the same time, the metal nanoparticles (Au, Pt, etc.) produced by metal electrode sputtering can be loaded on carbon nanospheres to obtain highly active carbon-supported noble metal nanoparticles electrocatalysts [35,36]. The carbon nanospheres obtained by this method exhibited a diameter range of 20 to 30 nm and a pore size range of 13 to 16 nm. The Au nanoparticles had a particle diameter of less than 10 nm and were uniformly dispersed on the carbon nanospheres (Figure 5c–e), making them an ideal carbon-supported, noble metal electrocatalyst.

Solution plasma process can also directly reduce noble metals to prepare electrocatalysts [37]. Lee et al. [38] synthesized Pt/C electrocatalysts with Pt nanoparticles being reduced by H_2_PtCl_6_·6H_2_O, while carbon supports were formed by the corrosion of carbon electrodes during the solution plasma process. The Pt nanoparticles with a diameter of about 38.14 nm were composed of many primary particles with a diameter of about 1.85 nm, which exposed more (111) crystal faces and exhibited good HOR catalytic activity. In 2018, Hussain et al. [39] used He/H_2_ plasma to reduce H_2_PtCl_6_ to Pt nanoparticles which were then loaded on nitrogen-doped reduced graphene oxide (rGO-N). The obtained Pt/rGO-N electrocatalysts showed a comparable high ORR electrocatalytic activity and superior stability to the commercial Pt/C. It was found that the ORR activity in the electrolytes of 0.1 M KOH and 0.05 M H_2_SO_4_ were three and two times higher than that in commercial Pt/C, respectively. Cui et al. [40] prepared PtO_a_PdO_b_@Ti_3_C_2_T*_x_* catalysts by loading PtPd bimetallic oxide nanoparticles onto two-dimensional MXene carrier by solution plasma reduction, as shown in Figure 6. The particle size of PtPb nanoparticles increased with the extension of plasma reaction time. The catalyst obtained by plasma reaction for 3 min showed good catalytic activity for HER and OER, exhibiting the activation potential of HER in 0.5 M H_2_SO_4_ of 57 mV, which was the same for that of OER in a 0.1 M KOH solution of 1.54 V at the current density of 10 mA/cm^2^. In particular, it had superior water electrolysis performance in an alkaline solution of 1.0 M KOH, which showed that the electrolysis water voltage at the current density of 10 mA/cm^2^ was 1.53 V.

### 2.4. Plasma Prepared and Modified Electrocatalysts Support

Plasma technology is also used to prepare and modify the catalyst carrier that achieves hydrophilicity and element doping, and promote the dispersion of the catalyst’s active components and the improvement of its catalytic activity [41,42,43]. Chetty et al. [44] formed abundant CO and COO-functional groups on the surface of multi-walled carbon nanotubes (CNT) by O_2_ plasma, which improved the dispersion of Pt-Ru nanoparticles and the catalytic activity toward MOR. Ding et al. [45] prepared N-doped graphene-coated Pt nanocrystals (N-GPN) by N_2_ plasma treatment of GPN (Figure 7a–f). From Figure 7g–h, it can be seen that nitrogen doping can produce more active sites and improve the electrocatalytic performance for ORR, which can effectively regulate the activity of the electrocatalysts. Loganathan et al. [46] prepared different Pt/C(N) and Pt/C(AA) catalysts with excellent ORR catalytic activities by modifying carbon black used for the fuel cell catalyst support system with N_2_ and allylamine plasma, respectively. Du et al. [47] realized in-situ N-doping and surface modification on carbon paper carrier surface by 25% N_2_ + 75% H_2_ plasma treatment, promoting the growth of Pt nanowire arrays (NW). The self-supporting Pt NW/GDL (large area gas diffusion layer) electrocatalysts with an area of 5 cm^2^, could be directly used as catalyst electrodes for fuel cells, which showed excellent charge transfer and mass transfer properties, and their power density was twice as high as that of traditional Pt nanoparticles.

In 2017, Hu et al. [48] obtained Pt/ZnO/KB photocatalysts with solution plasma technology. They first synthesized ZnO nanowires using zinc electrode wires, and then replaced them with Pt electrode wires, loaded Pt nanoparticles onto zinc oxide, and finally, placed Pt/ZnO in Keqin black suspension to obtain aa Pt/ZnO/KB photocatalyst. Pt/ZnO/KB exhibited excellent MOR photocatalytic activity due to the better electron transport performance and smaller resistance of ZnO nanowires. The catalytic activity of Pt/ZnO/KB was 964 mA/mg, which was more than three times that of Pt/KB (306 mA/mg). Meanwhile Pt/ZnO/KB showed excellent CO toxicity resistance and stability, which made it possible for methanol fuel cells to work all-weather in light or dark.

## 3. Non-Noble Metal Electrocatalysts

### 3.1. Metal and Alloy Electrocatalysts

In recent years, non-noble metal catalysts have attracted more and more attention. In particular, transition metals with empty orbits combined with molecules containing lone pairs of electrons through coordination bonds to form transition states with lower barriers reduce the activation energy of the whole reaction path and accelerate the chemical reaction. In 2016, Flis-Kabulska et al. [49] prepared Ni–Fe–C electrocatalysts by introducing carbon into NiFe alloy using CH_4_+H_2_ plasma carburization. Although the solubility of carbon in NiFe alloy was very low, a carbide layer with thickness of about 2 µm was formed on the surface, which greatly improved the surface hardness and corrosion resistance. Ni–Fe–C alloy exhibited good HER catalytic activity in 25 wt.% KOH solution at 80 °C. Jin et al. [50] used the microwave plasma chemical vapor deposition (MPECVD) method to coat carbon nanoparticles on carbon cloth, which prepared a three-dimensional CoNPs@C with excellent conductivity and catalytic activity. It was found that a low overpotential of 153 mV can be generated at the current density of 10 mA/cm^2^ for HER, and the overpotential was 270 mV for OER, and the decomposition voltage was 1.65 V when composing a full hydrolyzed cell.

Cu-based catalysts are commonly used in the preparation of hydrocarbons by CO_2_RR, such as carbon monoxide, methane, ethylene, ethanol, and n-propanol [51]. Mistry et al. [52] used plasma to treat oxidized Cu-based catalysts for the catalytic reduction of CO_2_ at low overpotential. It can be seen from Figure 8a–h that Cu_2_O was not completely reduced during the reaction, and Cu^+^ ions were still on the surface. Cu-based catalysts derived from oxides only had a minimal effect on catalytic performance, while Cu^+^ played an important role in reducing the initial potential and improving the selectivity of ethylene. Among the catalysts, Cu foil treated by O_2_ plasma had 60% ethylene selectivity at −0.9 V (versus reversible hydrogen electrode (RHE)) potential (Figure 8i,j). In 2017, Gao et al. [53] adjusted Cu (100) crystal face and the content of oxygen/chloride ion by treatment of Cu nanoparticles with H_2_, O_2_, and Ar plasma, respectively. It was found that the oxygen content on the surface and subsurface of Cu nanoparticles was the key to achieve high activity and selectivity of hydrocarbons/alcohols, even more important than the existence of the Cu (100) crystal plane. Cu nanoparticles treated by O_2_ plasma had lower overpotential and higher selectivity for ethylene, ethanol, and n-propanol, of which the selectivity for ethylene and ethanol was 45% and 22%, and the maximum Faraday efficiency (FE) of C2 and C3 products can reach 73%.

### 3.2. Transition Metal Oxides

#### 3.2.1. Cobalt Oxides

Cobalt oxide (CoO_x_) has attracted much attention for its superior catalytic activity in electrocatalysts [54,55]. In 2017, Kim et al. [56] prepared Ag@Co_3_O_4_ core-shell hybrid nanocrystals in aqueous solution by treating Ag and Co electrode wires with solution plasma technology. Compared with Ag and Ag–Co electrocatalysts prepared in alcohol solution, it showed a better electronic effect and geometric effect, which was beneficial to the fracture of the O–O bond for ORR, resulting in the catalytic activities being 5.2 and 2.6 times higher, respectively. The modification of the carrier can also effectively improve the catalytic activity. Taiwo et al. [57] synthesized three-dimensional nanoporous graphene sheets with highly specific surface areas as a catalyst carrier by microwave-induced plasma, and then doped them with nitrogen and supported Co_3_O_4_ nanoparticles to prepare 15Co_3_O_4_/N-AP/800 electrocatalysts. The catalysts exhibited excellent electrical conductivity and ORR performance, showing that the Tafel slope was 42 mV/dec, which was better than 82 mV/dec of Pt/C. In 2018, Lang et al. [58] used DC arc discharge plasma to synthesize CoO nanospheres mixed with La_2_O_3_ and Pt/C to prepare highly active CO–La–Pt ternary ORR catalysts for Li–O batteries. The specific capacity and energy density of the electrode reached 3250.2 mAh/g and 8574.2 Wh/kg at 0.025 mA/cm_2_, and the capacity retention rate reached 38.3% after 62 cycles.

Transition metal oxides with abundant defects and vacancies were prepared by plasma etching, which exhibited excellent electrocatalytic performance [59]. In 2016, Xu et al. [60] used Ar plasma to treat Co_3_O_4_ nanosheets, which produced oxygen vacancies on the surface of Co_3_O_4_ nanosheets, which had higher current density and lower initial potential than untreated Co_3_O_4_ nanosheets. The catalysts showed good OER performance and the current density was increased by 10 times at a voltage of 1.6 V. Wang et al. [61] introduced oxygen vacancies in cobalt oxyhydroxide (CoOOH) by Ar plasma technique. It was proven by density functional theory that Vo–COOH catalysts with oxygen vacancies could initiate additional reaction steps to accelerate the oxidation of H_2_O—H_2_O* ↔ (HO + H)* and (HO + H)* ↔ HO* + H^+^ + e^−^, and had a lower kinetic barrier. At the same time, the experimental observation confirmed that the two sites could promote the OER catalytic activity of V_0_–CoOOH, which accelerated the deprotonation process and enhanced water oxidation due to their synergistic catalysis. Meanwhile, this method can also be extended to other OER catalysts, which has broad application prospects. In 2018, Ma et al. [62] applied Co_3_O_4−*x*_ with oxygen-rich vacancies, treated by Ar plasma, to the catalytic reactions of ORR (half-wave potential 0.84 V) and OER (overpotential of 330 mV and Tafel slope of 58 mV/dec), resulting in excellent catalytic activity, as shown in Figure 9b. The Zn-based hybrid battery (Figure 9a) prepared on this basis can simultaneously carry out two kinds of electrochemical reactions (OER and ORR electrocatalysis, and reversible Co−O ↔ Co−O−OH redox reaction), with a high power density of 3200 W/kg and a high energy density of 1060 Wh/kg. The battery showed good water resistance and washing resistance. Its capacity retention rate was 99.2% after 20 h of immersion test and 93.2% after 1 h of washing test. More interestingly, when exposed to air from underwater, the battery can automatically restore the power output, with excellent electrochemical performance, good environmental adaptability, and “air recyclability,” so was considered to have potential to become the next generation of energy storage device and be widely used in flexible wearable electronic devices and other technical fields (Figure 9c).

The doping of heteroatoms can also improve the electrocatalytic activity of CoO_x_, and the plasma technology can be used to dope nitrogen [63]. In 2017, Xu et al. [64] found that surface etching and nitrogen doping were realized on Co_3_O_4_ nanosheets by N_2_ plasma. The sample N–Co_3_O_4_ had more active sites due to N-doping and oxygen vacancies, leading to a significant improvement in electronic conductivity. Compared to Co_3_O_4_ nanosheets (1.79 V and 234 mV/dec), it showed a lower potential (1.54 V) at a current density of 10 mA/cm^2^ and a Tafel slope of 59 mV/dec. Uhlig et al. [65] directly treated the CoAc/C precursor with N_2_ plasma, that simultaneously achieved nitrogen doping of the carrier and active components. The product showed excellent ORR catalytic activity in 0.1 M KOH and 0.1 M K_2_CO_3_ solutions. However, cobalt oxide produced more by-products, which may have resulted in membrane damage to the fuel cell, which remains unimproved upon.

Elemental P can be doped into an oxide by plasma [66]. In 2017, Xiao et al. [67] found that P atoms were immediately transported and filled into the position of an oxygen vacancy defect, which was a structural defect caused by surface oxygen etching of materials by Ar plasma airflow treatment (Figure 10a–f). The 3p orbital and 3d orbital of P atoms have lone pair of electrons, so the V_0_ of Co_3_O_4_ filled with P atoms can be used to effectively stabilize vacancies, which can induce local charge density and adjust surface charge state, and also can be used to adjust the relative proportion of Co^2+^/Co^3+^, improving the adsorption and electrocatalytic properties of materials [68]. As can be seen in Figure 10g–i, V_0_-Co_3_O_4_ had a relatively lower coordination number than Co_3_O_4_ due to the formation of oxygen vacancies, and P–Co_3_O_4_ had a much higher coordination number than V_0_–Co_3_O_4_ and Co_3_O_4_, due to the filling of P atoms. When P occupied the vacancy sites, electrons migrated out of the 3d orbital of Co, and more Co^2+^ (Td) remained in P–Co_3_O_4_, which exhibited excellent catalytic activity for HER and OER. P-Co_3_O_4_ required only an overpotential of 120 mV in the HER process and the overpotential of 280 mV in the OER process at the current density of 10 mA/cm^2^. The Tafel slopes for HER and OER were 52 mV/dec and 51.6 mV/dec, respectively. P–Co_3_O_4_ can effectively catalyze full water-splitting in a 5 M KOH solution (80 °C), which showed an overpotential of 420 mV at a current density of 100 mA/cm^2^.

In addition, excellent electrocatalytic performance can be obtained by reducing the size of the catalysts to expose more active sites. Dou et al. [69] prepared atomic-scale CoO_x_-ZIF (Zeolitic Imidazolate Framework, ZIF) catalysts by directly treating ZIF-67 precursors with O_2_ plasma. ZIF-67 has an abundant pore structure, which provided abundant channels for O_2_ to enter ZIF to activate Co^2+^ to obtain atomic-scale CoO_x_. After treatment with O_2_ plasma, the porous structure and large surface area of ZIF-67 were still retained, which were very conducive to the transport of materials toward OER. The prepared atomic-scale CoO_x_-ZIF catalysts can provide abundant active sites for oxygen evolution reaction, and its catalytic performance for OER was better than that of noble metal catalyst RuO_2_. The atomic scale electrocatalysts were obtained directly in-situ in MOFs (metal organic frameworks, MOFs) by that method for the first time, which provided a new idea for the convenient preparation of atomic scale electrocatalysts.

#### 3.2.2. Perovskite-Type Oxides

Perovskite-type oxide ABO_3_ is a new type of inorganic nonmetallic material with unique physical and chemical properties. A-site is usually the ion of a rare earth or alkaline earth element, and B-site is the ion of transition element. A-site and B-site can be partially replaced by other metal ions with similar radii and keep their crystal structures unchanged. Therefore, ABO_3_ is an ideal sample for studying the surface and catalytic performance of catalysts in theory, which has great development potential in the fields of environmental protection and industrial catalysis [70]. In particular, unlike the anion vacancies in other metal oxides, ABO_3_ can also achieve cation vacancies. Chen et al. [71] used Ar plasma technology to surface-treat the SnCo_0.9_Fe_0.1_(OH)_6_ (SnCoFe) precursor and adjusted the electrocatalytic activity of OER by defect engineering. Because Sn(OH)_4_ had weak chemical bonds and its lattice energy was much lower than that of cobalt or iron hydroxides, Sn vacancies preferentially formed in the process of forming various cationic vacancies, thus a perovskite-type hydroxide (SnCoFe–Ar) electrocatalyst with rich Sn vacancies has been developed. The abundant Sn vacancies made the catalyst materials have a larger specific surface area and superhydrophilicity, exposed more CoFe active sites, and created more 1–2 nm channels on the surfaces of the catalyst materials, which improved the material transport capacity and electronic transport capacity, and optimized the adsorption capacity of the reactants. Compared with the overpotential of 420 mV and Tafel slope of 77.0 mV/dec for SnCoFe, the overpotential and the Tafel slope of the SnCoFe–Ar were 300 mV and 42.3 mV/dec at the current density of 10 mA/cm^2^, which exhibited superior catalytic performance for OER.

Transition metal ions also affect the catalytic performance of the electrocatalysts. Hayden et al. [72] successfully prepared SrTi_1−x_Fe_x_O_3−y_ (STFO) perovskite-type composite gradient electrocatalysts by high-throughput physical vapor deposition (HT-PVD) with O_2_ plasma. With the increase of x value, the lattice parameters of STFO increased from 0.392 + 0.001 nm of SrTiO_3_ to 0.386 + 0.001 nm of SrFeO_3_, and the conductivity increased as well. When x > 0.75, the conductivity reached ρ = 0.041 S/cm. The corresponding OER reduced the initial potential at current 100 μA from 1.52 VRHE (x = 0.2) to 1.40 VRHE (x = 0.85), but the high OER activity was accompanied with low stability. Therefore, SrTi_0.5_Fe_0.5_O_3−y_ showed the best ORR activity and electrode stability.

In order to develop efficient electrocatalysts with extraordinary mass activity and stability, Chen et al. [73] deposited amorphous SrCo_0.85_Fe_0.1_P_0.05_O_3−δ_ (SCFP) nano-thin films with weak chemical bonds on the conductive nickel foam (NF) substrate by high-energy Ar plasma. The rapidly reconstituted SCFP-NF bifunctional catalyst exhibited good electron conductivity, ultra-high activity, and stability, due to the destruction of strong chemical bonds by plasma in the crystalline SCFP target. The ultra-high quality activity (overpotential 550 mV) of 1000 mA/mg in 1.0 M KOH solution was about 2.1 times higher than that of RuO_x_-NF coupled Pt-NF electrode, as shown in Figure 11a,b. At the same time, the catalysts exhibited outstanding catalytic activity with a water decomposition stability of 650 h (current density of 10 mA/cm^2^), which was significantly superior to the current RuO_x_-NF coupled Pt-NF electrode, which represented a major breakthrough in water cracking, as shown in Figure 11c–e. The simple reconstruction strategy was expected to be used in other advanced energy conversion and storage devices to develop new and efficient catalysts.

#### 3.2.3. Two-Dimensional (2D) Layered Double Hydroxides

Transition-metal layered double hydroxides (LDH) have unique 2D structures, large specific surface areas and special electronic structures, showing good electrocatalytic performance [74,75]. The LDHs prepared by the conventional methods have a high number of layers, which limits the exposure of the electrocatalytic active sites, and thus inhibits the electrocatalytic activity. At present, the method of liquid phase exfoliation to prepare LDHs generally uses organic solvents, which has the disadvantages of substantial time-consumption, toxicity, and easy adsorption of organic solvents. Compared with traditional liquid phase exfoliation, plasma exfoliation has the advantages of cleanliness, rapidity, and efficiency, and avoids the adsorption of organic solvent molecules. In 2017, Liu et al. [76] prepared 2D ultra-thin nanosheets (CoFe-LDHs–H_2_O) by treating bulk CoFe bimetallic hydroxide nanosheets (CoFe-LDHs) with water plasma technology, which can be stabilized in the form of powder, as shown in Figure 12a,b. Compared with bulk CoFe-LDHs, the thickness of CoFe-LDHs–H2O nanosheets decreased significantly (Figure 12c,g), and the 2D base surface became rough (Figure 12d,h). From Figure 12e,f,I,j, it can be seen that water plasma had a very good peeling effect on CoFe-LDHs nanosheets, as the thickness of nanosheets decreased from 25.2 nm to 1.54 nm. The obtained CoFe-LDHs–H_2_O had larger specific surface area, resulting in many cobalt vacancies, iron vacancies, and oxygen vacancies on its surface, which enhanced the adsorption of OER intermediates and further boosted the catalytic activity. Tafel slope and charge transfer impedance decreased significantly, such that the overpotential potential was only 36 mV/dec.

In the meantime, Wang et al. [77,78] obtained 2D ultra-thin CoFe-LDHs–Ar and CoFe-LDHs–N_2_ nanosheets by treating CoFe-LDHs with Ar plasma and N_2_ plasma, respectively. Gas plasma technology was not only non-toxic, clean and time-saving, but CoFe-LDHs–Ar and CoFe-LDHs–N_2_ exfoliated by the dry method also had abundant oxygen vacancies, cobalt vacancies, and iron vacancies. The formation of multiple vacancies was conducive to regulating the electronic structure of the material surface, reducing the coordination number around cobalt and iron atoms, increasing the degree of chaos around atoms, and facilitating the adsorption of the oxygen evolution reaction intermediate, which led to improving the electrocatalytic performance of the material. CoFe-LDHs–Ar nanosheets exfoliated by Ar plasma exhibited good oxygen precipitation performance. The overpotential of CoFe-LDHs–Ar nanosheets was only 266 mV at the current density of 10 mA/cm^2^, showing good activity. CoFe-LDHs–N_2_ nanosheets stripped by N_2_ plasma not only had abundant metal vacancies and oxygen vacancies, but also realized nitrogen doping. The doping of elemental nitrogen was beneficial to changing the electronic structure around the reaction site and improving the catalytic activity by adsorbing oxygen precipitation intermediates, and the resulting defect-rich 2D ultra-thin CoFe-LDHs–N_2_ nanosheets made it easy to expose more electrocatalytic active sites to increase their oxygen evolution performance. The overpotential was only 233 mV at the current density of 10 mA/cm^2^, and the Tafel slope and charge transfer impedance were also reduced, which had very good stability. The new method of stripping 2D layered materials to achieve nitrogen doping and defect-richness simultaneously can be used for reference to other similar materials.

#### 3.2.4. Other Metal Oxides

In addition, other transition metal oxides are also used as electrocatalysts. For example, in 2017, Oturan et al. [79] prepared sub-stoichiometric titanium oxide (Ti_4_O_7_) by plasma deposition and electrocatalytic oxidation to treat the pollutant antibiotic amoxicillin (AMX), which can rapidly oxidize and degrade 0.1 mM (36.5 mg/L) AMX in a short time. Luo et al. [80] prepared FeO_x_/C electrocatalysts by uniformly depositing iron oxide clusters on porous carbon substrates with high ionized iron plasma generated by arc discharge. It was found that the catalysts showed good ORR and OER properties, and an excellent rate performance and cycle life in a Li-O_2_ battery. A discharge capacity of 500 mAh/g was retained after 37 cycles at the current density of 100 mA/cm^2^. In 2018, Peng et al. [81] prepared MnO_x_@C-D electrocatalysts by dielectric barrier discharge technology for ORR. Because of the effect of plasma, the catalysts allowed manganese ions with different valences to coexist, and had high oxygen adsorption capacities. The aggregation of nanometer catalysts was inhibited at a medium temperature, which showed a higher reaction activity than that of the catalysts prepared by traditional calcination method. Jiang et al. [82] induced abundant defects on the surface of MnO_2_ nanowires by Ar plasma etching, and observed that Ar–MnO_2_ had abundant edges and oxygen vacancies to produce more active sites, showing excellent ORR performance. At the current density of 157 mA/cm^2^, the power density of Al-air battery based on A–MnO_2_ catalysts can reach 159 mW/cm^2^, which is much higher than that of untreated MnO_2_ catalysts (115 mW/cm^2^).

In the process of HER catalysis, pre-reduction of transition metal oxides is an effective way to improve the catalytic activity of HER. Zhang et al. [83] treated NiMoO_4_ nanowire arrays by carbon plasma for 30 s, which formed Ni_4_Mo nanoclusters on the surface and deposited a layer of graphitized carbon, as shown in Figure 13a–d. Compared with NiMoO_4_ nanowire arrays treated by H_2_ reduction, the sample (C-60s) exhibited excellent catalytic activity for hydrogen evolution (Figure 13e–g), and could maintain its array morphology, chemical composition, and catalytic activity during long-term intermittent hydrogen evolution. It opened up a new way for simultaneous activation and stabilization of transition metal oxide electrocatalysts. In 2018, Geng et al. [84] used H_2_ plasma to surface-treatment of 2D ZnO nanosheets, which produced abundant oxygen vacancy defects on the surface. Although these oxygen vacancy defects did not increase the number of active centers, they can lead to the efficient activation of CO_2_ molecules in the electron-rich state and enhance the electrochemical catalytic reduction activity of CO_2_. At the voltage of −1.1 V compared to RHE, the oxygen-rich ZnO nanosheets with vacancies can effectively reduce the CO_2_ to a CO product with a current density of −16.1 mA/cm^2^ and a Faraday efficiency of about 83%, exhibiting excellent catalytic activity for CO_2_RR.

### 3.3. Transition Metal Sulfides

Transition metal sulfides which have received extensive attention in recent years, have become new and highly efficient electrocatalysts, particularly HER catalysts. The hydrogen evolution reaction mainly occurs at the boundary of 2D materials in which the atom is incompletely coordinated. Therefore, the preparation of 2D transition metal sulfides with incompletely coordinated atomic boundaries is the key to whether the material can replace the noble metal platinum. Li et al. [85] etched 2D TaS_2_ nanosheets by using O_2_ plasma technology. It was found that the plasma could control the processing of high-density atomic-scale pores on three-dimensional (3D) two-layer crystals, which can increase the catalytic sites and effectively improve the hydrogen evolution catalytic activity of 2D materials, as shown in Figure 14a. As can be seen from Figure 14b–g, the atomic defect density increased with the increase of processing time, which exhibited excellent HER catalytic activity of treatment at 15 min.

2D MoS_2_ is also used for electrocatalytic reactions, which can also be used to enhance the activity by Ar, O_2_, and H_2_ plasma. Tao et al. [86] prepared Ar–MoS_2_ and O_2_–MoS_2_ thin film materials for OER catalytic reaction using Ar and O_2_ plasma treatment, respectively. The hydrophilic contact angles of Ar and O_2_–MoS_2_ were found to be 61.8° and 48.4°, which were significantly better than untreated MoS_2_ (96.5°). The Tafel slope of O_2_–MoS_2_ was up to 105 mV/dec, which was significantly better than Ar–MoS_2_ (117 mV/dec) and MoS_2_ (160 mV/dec). In 2018, Zhang et al. [87] and Huang et al. [88] found that MoO_3_ species were formed at the edge and plane positions of MoS_2_ by using O_2_ plasma to treat MoS_2_ and H_3_Mo_12_O_40_P/MoS_2_ composite intercalation compounds respectively. The MoO_3_ can also be reduced and decomposed from the lattice of MoS_2_ in the HER catalytic process, effectively improving the HER catalytic activity of MoS_2_. In addition, the presence of S defects also enhanced the catalytic activity of MoS_2_ for HER. Cheng et al. [89] treated MoS_2_ with H_2_ plasma, and the sample MoS_2_ treated for 15 min produced high density S vacancies on the base plane of single-layer crystalline MoS_2_, whose overpotential could reach 240 mV (versus RHE) at the current density of 10 mA/cm^2^. The amorphous a–MoS_2_ treated by H_2_ plasma had abundant sulfur vacancies, resulting in a decrease of overpotential from 206 mV of a–MoS_2_ to 143 mV of MoS_1.7_ [90].

Cobalt sulfide also has good electrocatalytic activity. Dou et al. [91] prepared bifunctional catalysts with catalytic ORR and OER by simultaneously etching and doping the catalyst’s surface and using NH_3_ plasma. Co_9_S_8_ nanoparticles (Co_9_S_8_/G) loaded on graphene were treated by ammonia plasma. Nitrogen was successfully doped into the lattices of Co_9_S_8_ and graphene by the treatment of Co_9_S_8_ nanoparticles loaded on graphene (Co_9_S_8_/G) with NH_3_ plasma. Moreover, partial etching also occurred on the surface of Co_9_S_8_ and graphene. The doping of heteroatoms can effectively adjust the electronic structure of Co_9_S_8_ and graphene, while the etching of the surface can expose the catalysts to more catalytically active sites, which can obtain a high activity bifunctional electrocatalysts with similar ORR catalytic performance to commercial Pt/C, and OER catalytic performance superior to RuO_2_. In 2018, Zhang et al. [92] prepared a sulfur-rich, Co_3_S_4_, ultrathin porous nanosheet with abundant sulfur vacancies (Co_3_S_4_ PNS_vac_) using Ar plasma to treat the Co_3_S_4_/TETA precursor. When the hydrogen evolution potential was 200 mV, the mass activity was as high as 1056.6 A/g, which is 107 and 14 times that of Co_3_S_4_ nanoparticles and nanosheets respectively, which was also better than the HER electrocatalytic activity of Pt/C. In addition, the simple and rapid chemical conversion strategy can be extended to the synthesis of ultra-thin porous CoSe_2_ and NiSe_2_ sheets with anion-rich defects, due to their universality, which opened up new ways to control the electrocatalytic performance of 2D nanomaterials through defect engineering and ultra-thin porous structure engineering.

In recent years, self-supporting electrode materials have attracted widespread attention. In 2018, He et al. [93] synthesized CuI nanosheet arrays by using dielectric barrier discharge (DBD) plasma with iodine vapor on copper foam, and then prepared Cu_2_S nanosheet arrays via sulfur ion exchange. The Cu_2_S/CF showed excellent OER catalytic activity in 1 M KOH solution with an overpotential of only 290 mV at the current density of 10 mA/cm^2^. Yeo et al. [94] used a PEALD technique to grow a tungsten disulfide (WS_2_) film on a Ni mesh (WS_2_/Ni-foam), which exhibited superior HER activity. In acidic electrolyte, the overpotential and the Tafel slope of the catalysts were 280 mV and 63 mV/dec at a high working current density of 100 mA/cm^2^. Daniel et al. [95] modified WS_2_ and MoS_2_ by SF_6_/C_4_F_8_ plasma. It was found that their overpotentials shifted to 100 mV and 200 mV, and the Tafel slopes decreased by 50 mV/dec and 120 mV/dec, respectively. Especially for WS_2_ samples treated by plasma for 31s, the Tafel slope was only 81 mV/dec. Plasma can also be used to treat the carrier. Qu et al. [96] used a plasma-treated Ni–Fe foam (PNFF) carrier to form a PNFF with many micro-grooves on the surface. Then, CoS/Ni_3_S_2_–FeS nanoflowers were formed by vulcanization, after Co nanoparticles were deposited on PNFF carrier by electrodeposition, which showed the overpotentials toward HER and OER of 75 mV and 136 mV at the current density of 10 mA/cm^2^, respectively. The amount of H_2_ and O_2_ produced were 680 μmol/h and 1230 μmol/h, respectively, showing high electrocatalytic activity and overall water splitting performance.

### 3.4. Transition Metal Selenides

Transition metal selenides also exhibit excellent electrocatalytic performance, especially for the HER catalytic reaction. [97] HER has become the bottleneck of hydrogen production from electrolytic water due to the slow reaction kinetics. The lower overpotential and the Tafel slope identify the benefits of HER catalytic activity; plus the HER can react at a lower applied voltage. For example, the Tafel slope of a Pt-based catalysts can be reached 30 mV/dec, which is close to the theoretical limit of 29 mV/dec, but its reserves and price limit wider applications. In 2017, Deng et al. [98] used a 3D porous vertical graphene array (VG) prepared by MPECVD as a conductive substrate, and prepared a MoSe_2_/VG/array by hydrothermal method treated by ammonia gas heat treatment modification. Next, the 2H to 1T (2H-1T) phase transition of MoSe_2_ was initiated by N doping to form N–MoSe_2_/VG having a 2H-1T composite phase. The introduction of 1T phase reduced the band width of MoSe_2_ and improved the electron transport performance. At the same time, N doping increased the hydrogen evolution active site at the edge of MoSe_2_ sheet, which showed that the Tafel slope can reach 49 mV/dec, demonstrating a good hydrogen evolution performance. Qu et al. [99] used the glangcing angle deposition (GLAD) method combined with the N_2_/H_2_ plasma technology to prepare the MoSe_2_/Mo composite, as shown in Figure 15. The plasma-assisted selenization process gave MoSe_2_/Mo a large surface area, while accelerating the charge transfer of the metal phase of MoS_2_ exposed to the edge. The Tafel slope of catalysts can reach 34.7 mV/dec, which exhibits excellent HER catalytic activity.

### 3.5. Transition Metal Nitrides

Transition metal nitrides are often used as electrocatalysts, known as “quasi-platinum catalysts,” providing a new way to prepare economical and efficient electrocatalysts. Zhang et al. [100] prepared a three-dimensional porous NiMoN material on carbon cloth with N_2_ plasma treatment technology, which displayed high roughness and electron transport capability. The overpotential can reach 109 mV at a current density of 10 mA/cm^2^ due to its excellent synergistic effects from Ni, Mo, and N for HER performance. Ouyang et al. [101] used N_2_ plasma treatment to grow a 3D structure of nitriding (hNi_3_N) on Ni foam (NF), which showed excellent OER catalytic activity and laid the foundation for the development of a high-performance, metal nitride, energy storage switching electrode. In 2018, Liu et al. [102] produced nickel nitride (Ni_3_N_1−x_) rich in nitrogen vacancies by nitriding Ni foam (NF) by microwave plasma-generation (Figure 16a,b). The presence of nitrogen vacancies effectively promoted the adsorption of water molecules, improved the adsorption-desorption behavior of the intermediate adsorbed hydrogen, and enhanced the HER activity of Ni_3_N_1−x_, as shown in Figure 16c–f. The self-supporting Ni_3_N_1−x_/NF electrode showed an overpotential of 55 mV and a Tafel slope of 54 mV/dec in an alkaline solution at the current density of 10 mA/cm^2^.

Some nitride also exhibits excellent electrocatalytic properties in terms of ORR catalytic performance. Wang et al. [103] deposited high-density discrete Cu_3_N nanocrystals on XC-72 carbon black by plasma enhanced atomic layer deposition (PEALD). It was found that the work function of Cu_3_N nanocrystals was 5.04 eV, which was lower than Pt (5.60 eV). Cu_3_N had stronger electron transfer performance than typical Pt catalysts. At the same time, the synergistic coupling effect between Cu_3_N nanocrystals and carbon support made the Cu_3_N_200_/C sample exhibit a smaller π (= 4.34 eV) than the pure Cu_3_N nanocrystals. It displayed excellent ORR catalytic activity, significantly improved quality activity, and greater durability. Panomsuwan et al. [104] prepared iron-nitrogen-doped carbon nanoparticle–carbon fiber (Fe–N–CNP–CNF) by solution plasma. Because of the synergistic effect of the high graphitization of CNF, the mesoporous/macroporous CNP, the catalytic active center of ORR (graphite N and Fe-N bond), and the carbon-encapsulated Fe/Fe_3_C particles, the Fe–N–CNP–CNF obtained showed excellent catalytic activity, durability and methanol resistance toward oxygen reduction (ORR) in an alkaline solution. Zhong et al. [105] found that the use of air plasma etching of iron-nitrogen co-doped porous carbon (Fe–N/C) electrocatalysts can remove sp^3^ Cs with poor stability and amorphous sp^2^ Cs, exposing more active catalytic FeN_4_ centers, and transforming a small number of Fe-based nanoparticles into FeN_4_ species, which significantly improved the catalytic activity of Fe–N/C for the oxygen reduction reaction (ORR) in acidic and alkaline electrolytes.

### 3.6. Transition Metal Phosphides

Transition metal phosphides have attracted great interest as electrocatalytic catalysts, particularly for HER and OER catalysts for electrocatalytic water splitting [106]. Currently, bimetallic phosphides exhibit higher catalytic activity than monometallic compounds [107] Liang et al. [108] realized the transition of NiCo–OH to NiCoP on nickel foam by PH_3_ plasma, and prepared NiCoP electrocatalysts, as shown in Figure 17a, which exhibited excellent HER and OER electrocatalytic activity under alkaline conditions (Figure 17b–d). Zhang et al. [109] used plasma-enhanced chemical vapor deposition (PECVD) to form phosphate and phosphide groups on foamed nickel in the presence of PH_3_, CO_2_, and H_2_ to form NiFePi/P. The change in the surrounding electronic environment of metal ions, due to the strong synergistic effect between phosphate and phosphide, not only increased the active center, but also improved the wettability and metal properties of the catalyst, the high conductivity, the wettability, and the active sites, which lead to excellent OER performance in an alkaline solution.

The doping of metal elements can also effectively balance the adsorption/desorption of oxygen and hydrogen-containing intermediates of metal phosphides, while enhancing density of electronic states’ (DOS) strength and enhancing charge transfer kinetics. Dinh et al. [110] etched metal V-doped Ni_2_P nanosheets (V-Ni_2_P) by O_2_ plasma, and found that V substitution in the prepared OV–Ni_2_P may introduce defects and crystal dislocations, thus increasing the number of active sites and the carrier concentration. The strength of the oxygen bonding and the charge transfer kinetics were significantly enhanced after vanadium doping. Due to the O_2_ plasma treatment, the phosphide surface was partially oxidized to produce a phosphate–phosphide region which balanced the adsorption/desorption of hydrogen and oxygen-containing intermediates. After the plasma, the Brunauer-Emmett-Teller (BET) surface area and the hydrophilicity of the material increased significantly. O–V–Ni_2_P had a large surface area (168.2 m^2^/g) and excellent hydrophilicity (contact angle of 16.8°), leading to a higher electroactive surface area and a lower charge transfer resistance. The study found that the 10% V-doped Ni_2_P nanosheets (O_3_–V_10_–Ni_2_P) treated by O_2_ plasma for 3 min gave the best performance. In the 1.0 M KOH solution, η10-HER and η10-OER were 108 mV and 257 mV, respectively. In addition, a smaller Tafel slope was obtained for HER (72.3 mV/dec) and OER (43.5 mV/dec). As comprehensive, dual-function catalysts for water electrolysis, O_3_–V_10_–Ni_2_P||O_3_–V_10_–Ni_2_P batteries required η10 to be only 1.563 V, which was superior to the most advanced IrO_2_||Pt/C (1.687 V). The catalysts also had significant durability and 20 h of operational stability. These excellent properties were attributed to high-valence vanadium doping, which acted as a good electron acceptor and contributed to OER performance. Peng et al. [111] prepared nickel-doped amorphous FeP nanoparticle-supported titanium nitride nanowire array (Ni–FeP/TiN/CC) composites by high-energy metal ion implantation, which showed excellent HER performance in alkaline systems. The overpotential was 75 mV at a current density of 10 mA/cm^2^, and the value remained at 93% of the initial value after continuous hydrogen evolution for 10 h at a high overpotential of 300 mV.

Cobalt phosphide is a new type of catalyst that has been developed in recent years to replace platinum metal-based electrocatalysts. However, the stability of cobalt phosphide is not as good as platinum, which limits the large-scale application of CoP_x_ in water electrolyzers. Goryachev et al. [112] prepared Co_3_O_4_ films by plasma enhanced atomic layer deposition method, and obtained CoP_x_ electrodes by thermal phosphating (PH_3_), which has CoP and surface rich P (P/CO>1). It was found that the surface activation energy and the exchange current density was 81 ± 15 kJ/mol and j_0_ = −8.9 × 105 A/cm^2^. Liu et al. [113] obtained plasma-activated (PA)–CoPO by the treatment of Co_3_(PO_4_)_2_ nanosheet arrays on a nickel network with H_2_ plasma, which had larger surface area, enhanced conductivity, rich coordination unsaturated Co^3+^, and numerous oxygen vacancies. The catalysts showed excellent OER and HER performance. The voltages of oxygen evolution and hydrogen evolution were 240 mV and 50 mV at a current density of 10 mA/cm^2^, and the Tafel slopes were 53 mV/dec and 35 mV/dec, respectively. Using a PA–CoPO nanosheet array as the anode and cathode, a full water split of 10 mA/cm^2^ was achieved at a low voltage of 1.48 V, which was superior to IrO_2_/C–Pt/C at a sufficiently high overpotential. That provides a path for the design of high performance electrocatalysts for total hydrolysis, as shown in Figure 17e,f.

### 3.7. Transition Metal Carbides

Nickel carbide (Ni_3_C_x_) films are prepared by the H_2_ plasma atomic layer deposition technique. They are polycrystalline and highly homogeneous, with a rhombic Ni_3_C crystal structure, and without any nanographite or amorphous carbon [114]. Xiong et al. [115] used the process to conformally coat a uniform thin layer of Ni_3_C on carbon nanotubes (CNTs), in order to obtain core-shell nanostructured Ni_3_C/CNT composites. The ALD-prepared Ni_3_C/CNT composite exhibited excellent performance in supercapacitors and electrocatalytic hydrogen evolution. Ko et al. [116] used plasma-assisted deposition to grow WC nanowalls from bottom to top on a silicon wafer, which were highly crystalline and showed superior HER performance in a 0.5 M H_2_SO_4_ solution. Not only was the Tafel slope 67 mV/dec, but also oxidation would not occur even after 10,000 cycles of recycling, which displayed good durability.

### 3.8. Other Compounds

In 2018, Guo et al. [117] performed a non-destructive modification of the Prussian blue analog (PBA) structure by air plasma. The reactive oxygen species produced by plasma selectively bound to the metal in the framework, while retaining the porous structure of the framework with the high dispersion and orderliness of metal sites in the framework. Porous catalyst Co–PBA-plasma 2 h was obtained with extremely high oxygen evolution activity. Prussian blue was a Fe/Co double metal cyanide skeleton composed of cyanide bridged Fe and Co cations (Figure 18a–c). The reactive oxygen species generated in the plasma were bonded to the open sites of Co, promoting the oxidation state transition of Co to Co(III) (Figure 18d). Due to the highly reactive metal catalytic sites in the nanoporous framework, the OER catalysts showed a low overpotential characteristic of only 330 mV at a high current density of 100 mA/cm^2^. This value is close to the actual operating conditions of industrial alkaline materials, implying a very promising technology for developing high performance catalysts. Yan et al. [118] used Ar plasma to prepare phytic acid–Co^2+^ (P-Phy–Co^2+^) with coordination unsaturation, which had an oxygen evolution potential of 306 mV at a current density of 10 mA/cm^2^. The method can be extended to CoFe bimetallic system. The oxygen evolution potential of P-Phy–Co^2+^/Fe^3+^ prepared at the current density of 10 mA/cm^2^ was 265 mV, and the Tafel slope was less than 36.51 mV/dec.

## 4. Carbon Based Electrocatalysts

Carbon is one of the most abundant and important elements in nature. Carbon-based catalysts have become one of the most popular electrocatalysts in recent years [119]. Carbon-based materials have more advantages than traditional materials, such as low costs, various structures, and good electrical and thermal conductivity. The excellent catalytic performance can be obtained by tailoring carbon-based materials with specific sizes, doping types, contents, morphologies and structures, promoting active site exposure, increasing the transport of reaction-related substances, and enhancing the transfer of electrons throughout the electrode [120,121]. At the same time, more defects become the active sites of the electrocatalyst after heteroatom doping. In addition, intrinsically defective carbon electrode catalysts also exhibit catalytic activity comparable to carbon materials doped with heteroatoms (e.g., F, S, P, and B) [122].

### 4.1. Defective Carbon Materials

Defective carbon materials have also attracted attention for enhancing electrocatalytic activity. Generally, defect-rich materials are prepared at elevated temperatures or using template precursors to form more edge and hole defects [123]. In 2016, Tao et al. [124] obtained defect-rich graphene and carbon nanotubes by using Ar plasma to etch the surface of graphene and carbon nanotubes, as shown in Figure 19a–e. The P-CNTs acquired with plasma treatment had also many defects.

Subsequently, Liu et al. [125] directly treated commercial carbon cloth with Ar plasma technology and exposed it to air, which not only made the surface of carbon fiber more rough and porous, but also exposed more graphene-like nanosheets on the surface. And the deficient, oxygen-doped graphene was also produced in situ on its surface. Compared with pure carbon cloth, the distance between carbon nanosheets was about 0.37 nm after simple plasma etching of carbon cloth P-CC, which was larger than that of graphite d_002_ = 0.34 nm. The value of SP^2^/SP^3^ of carbon was significantly reduced, which means that plasma treatment can produce more defects. The carbon fibers after plasma etching exhibited a larger specific surface area, exposing more active sites, and the treated carbon fibers had better conductivity, making them more conducive to material transfer, resulting in better catalytic performance in OER and ORR. An amorphous, edge-rich/defective graphene was generated in situ on the surface of the carbon fibers by argon plasma etching, while the dangling bonds of these defect sites were exposed to air and reacted with oxygen or water to achieve oxygen functionalization. In 2018, Lehmann et al. [126] used plasma-enhanced chemical vapor deposition to prepare defect-rich layered carbon nanowalls (hCNW) with readily accessible graphite edge locations at the top of the wall and many defect locations within the porous sidewalls, which was considered to be the ideal sites for adsorption and electron transfer. And the ORR initiation potential of hCNW-60 in 0.1 M KOH solution was 830 mV with a two electron transfer.

### 4.2. Nitrogen-Doped Carbon Materials

The doping of nitrogen atoms in graphene is considered to be a good way to improve the electrocatalytic performance. Wang et al. [127] mainly prepared N-doped graphene oxide (N–PEGO) by low temperature plasma technology. Ammonium carbonate was used as the activator and nitrogen source in the preparation of N–PEGO by low temperature plasma technology, which can effectively achieve the exfoliation of graphene oxide and nitrogen doping in one step (Figure 20). The N–PEGO prepared by this method has a nitrogen doping amount of 5.3 at%, a specific surface area of 380.0 m^2^/g, an initial potential of 0.89 V (versus RHE), and an excellent oxygen reduction reaction (ORR) catalysis, which also showed better stability and methanol resistance than the commercial noble metal Pt/C catalysts. Zhu et al. [128] achieved nitrogen doping of graphene foam by N_2_ plasma technology (NGF–CFP), which exhibited excellent OER performance.

N-doped carbon nanotubes are also used as electrocatalysts [129]. In 2015, Du et al. [130] successfully synthesized nitrogen-doped carbon nanotubes (NCNTs) with a nitrogen content of 5.38 at.% via microwave plasma chemical vapor deposition (MPCVD). Because of the doping of nitrogen, NCNT exhibited the same ORR electrocatalytic activity as Pt–CNT, which was 0.87 V of the initial point and 4.1 of the electron transfer number, and had the highest ORR performance at a load of 729 μg/cm^2^. Subramanian et al. [131] prepared carbon nanotube carpets (VA-NCNTs) with nitrogen-doped vertical alignment by treating carbon nanotubes with N_2_ plasma. Compared with undoped nitrogen VA-CNTs, VA-NCNTs exhibited well electrocatalytic performance under alkaline conditions. Zhang et al. [132] obtained NCNT/glass carbon (GC) electrode by treating carbon nanotubes with NH_3_ plasma, which can reduce CO_2_ to formate in water without using metal catalysts. If polyethyleneimine (PEI) is coated on NCNT/GC electrode, the catalytic overpotential can be significantly reduced, and the current density and efficiency can be increased, which is helpful for stabilizing the intermediate of the CO_2_ reduction of PEI.

The plasma deposition method and the solution plasma method are considered to be effective methods for one-step synthesis of nitrogen-doped carbon nanoparticles (NCNP) [133,134]. NCNPs can be synthesized in situ by solution plasma, and the type of C–N bond can be controlled by the structure of precursors and additives (Figure 21a–c). Li et al. [135] prepared different NCNPs using pyridine and acrylonitrile as heterocyclic and linear structure precursors, and using ruthenium as additives, which can realize the control of nitrogen elements (Figure 21d–g). It was found that the current density was proportional to the content of graphite-N. The existence of graphite-N promoted the direct four-electron transfer pathway of ORR, while the higher percentage of amino-N made the ORR initiation potential move to a positive value. Amino-N and graphite-N played a synergistic role in improving ORR activity. Li et al. [136] produced N-doped carbon nanoparticles by solution plasma containing pyridine-N, amino-N, and graphite-N bonds, which provided a simple and effective method to study the relationship between C–N bonding structure and the electrochemical performance of N-doped carbon catalysts. At the same time, Panomsuwan et al. [137,138] obtained NCNPs samples with different nitrogen doping content (0.63–1.94 at.%) by changing the difference of C/N molar ratio in organic precursors, using organic liquid mixtures such as benzene and pyrazine as precursors. It was found that the initial potential and current density of the ORR electrochemical properties of NCNP were improved with the increase of nitrogen doping content, which was mainly due to the graphite-N and pyridine-N on NCNPs. Compared to commercial Pt/C catalysts, NCNP showed superior long-term durability and strong methanol tolerance.

### 4.3. Oxygen-Doped Carbon Materials

The solution plasma method can also be used to synthesize carbon nanoparticles containing oxygen in one step. Ishizaki et al. [140] successfully synthesized oxygen-containing carbon nanomaterials using a mixture of benzene (BZ) and 1,4-dioxane (DO). Although DO content did not affect the initial potential, it did influence the current density of ORR. With the increase of DO content, the O contents in the samples increased, such that the order of current density of ORR carbon nanosamples was as follows: BZ90 + DO10 > BZ100 > BZ70 + DO30 > BZ50 + DO50. Kondratowicz et al. [141] used oxygen plasma to adjust the properties of reduced graphene oxide (rGO), which was an easy-to-control and eco-friendly method. Oxygen plasma treatment can improve the adsorption of enzymes on rGO electrodes by introducing oxygen groups and increasing porosity. With different plasma treatment times, different oxygen groups (such as carboxyl and hydroxyl groups) can be introduced on the surface of rGO to change the wettability of rGO, and other functional groups (such as quinones and lactones) can also be produced in a longer treatment time. In addition, the external surface of rGO was partially etched, resulting in an increase in surface area and porosity of the material. The current density of rGO treated for 10 min was twice as high as that of untreated rGO.

### 4.4. Sulfur-Doped Carbon Materials

Sulphur atom doping can effectively change the electronic and chemical properties of graphene, making it possible for graphene to be used as the electrocatalyst. In 2016, Wang et al. [142] realized the reduction of graphene oxide and doping of S with the treatment of microwave-assisted stripping and hydrogen sulfide plasma. Sulfur-doped graphene was used for an ORR electrocatalytic reaction under alkaline conditions, showing excellent electrochemical performance. In 2017, Ting et al. [143] etched sulfur-doped graphene by Ar plasma, resulting in more topological defects, while maintaining the original doping structure of SG (Figure 22a). Benefiting from the synergistic coupling of S-doping and plasma-induced topographic defects, SG–P had greatly enhanced HER activity and good stability in acidic media, demonstrated by its low overpotential of 178 mVat the current density of 10 mA/cm^2^, and Tafel slope of 86 mV/dec. The optimum HER activity of SG–P can be obtained by combining thiophene-rich substances with appropriate plasma-induced topological effects (Figure 22b–g). In 2018, Zhou et al. [144] obtained the self-supporting electrode 3DSG-Ar by Ar plasma treatment of three-dimensional sulfur-doped graphene (3DSG), which showed excellent HER electrocatalytic activity. After 2000 cycles, it still had good electrocatalytic stability, and the Tafel slope was 64 mV/dec.

### 4.5. Boron-Doped Carbon Materials

Panomsuwan et al. [145] prepared boron-doped carbon nanoparticles (BCNP) by solution plasma process using benzene and triphenyl borate as precursors (Figure 23). Compared with undoped carbon nanoparticles, the electrocatalytic activity of BCNP for oxygen reduction reaction (ORR) in alkaline solution was improved in terms of initial potential and current density. In addition, BCNP showed excellent long-term durability and methanol oxidation resistance in ORR. Li et al. [146] obtained porous BDD/Ta multilayers by etching polycrystalline boron-doped diamond (BDD) on tantalum substrates with H_2_/Ar plasma. It was found that the effective electroactive surface area and charge transfer ability of etched microcrystals at liquid crystal interface were improved. The porous BDD/TA electrodes were applied to electro-Fenton method for rapid degradation of methylene blue.

### 4.6. Fluorine-Doped Carbon Materials

Fluorine-doped carbon nanoparticles (FCNPs) can also be prepared by the solution plasma method. Panomsuwan et al. [147] used the mixture of toluene and trifluorotoluene as the precursor to prepare FCNPs, whose fluorine doping content can range from 0.95 to 4.52 at%. The obtained FCNPs mainly exhibited disordered amorphous structure, and the incorporation of fluorine atoms resulted in more defect sites and disordered structures in carbon particles. With the increase of fluorine doping content, the ORR electrocatalytic activity of FCNPs had been significantly improved, which was mainly due to the intercalation of ionic C–F and semi-ionic C–F bonds in the carbon structure. Compared with commercial Pt-based catalysts, FCNP exhibited excellent long-term operational durability and strong tolerance to methanol oxidation.

### 4.7. Heteroatom Co-Doped Carbon Materials

Diatomic co-doping is also considered as one of the effective strategies to improve the catalytic activity of carbon-based electrocatalysts. Tian et al. [148] obtained defect-rich P–NSG samples by the etching of N and S co-doped graphene (NSG) with Ar plasma. The synergistic coupling of N and S co-doping and plasma-induced structural defects maximized the number of active sites of graphene, which significantly improved the HER catalytic activity of P–NSG in both acidic and alkaline media. Lee et al. [149] synthesized boron–carbon–nitrogen (BCN) nanocarbon materials by solution plasma method in one step. The synergistic effects of N and B at the state of uncoupled bonding changed the electronic structure of basic carbon and promoted the formation of new ORR active sites. Although the electron transfer number of BCN nanocarbon was 3.43, and the ORR activity was not as good as commercial Pt/C, the current only decreased by 15.1% after 20,000 s, which was obviously better than commercial Pt/C (under the same conditions, the current decreased by 61.5%).

## 5. Conclusions and Prospects

Electrocatalytic materials have been widely used in energy and environmental fields, such as electrocatalytic hydrogen production, the reduction of carbon dioxide, fuel cells, the N_2_ reduction of ammonia, and so on. Noble metals have high energy utilization efficiency and excellent catalytic performance, but they have the disadvantages of poor availability and high prices. Therefore, it is important to develop highly active metal catalysts with low mass loading and high dispersibility. In recent years, researchers have focused on non-noble metal-based catalysts with low costs, high catalytic activities and long lives, such as transition metal catalysts, oxides, sulfides, nitrides, phosphides, and carbides. At the same time, carbon-based catalysts have been favored by researchers because of their superior electrocatalytic performances in both acidic and alkaline systems. The plasma device has a simple and adjustable structure, which can be used to prepare and modify the electrocatalysts.

In the preparation of electrocatalysts by plasma, different types of plasma may have different effects on the materials. The plasma can assist the precursor of gas or liquid-derived atoms, molecules, ions, or radicals to drive the synthesis of electrocatalysts. For example, the plasma deposition method can directly prepare a catalyst material with low loading and excellent dispersibility; the solution plasma can directly synthesize noble metal catalysts, noble metal alloy catalysts, and hetero atom-doped carbon-based catalysts; as a “bottom-up” synthesis technology, low-temperature plasma can be directly used to prepare specific nanostructured electrocatalysts. However, how to further improve the efficiency of plasma-assisted synthesis and expand its application in large-scale industrial production, is still challenging.

Plasma can achieve “top-down” denudation or surface treatment in plasma modification of electrocatalysts. Since the electrocatalytic process often occurs on the surface of catalysts, the structure regulation of the surfaces of catalysts by plasma can effectively improve the catalytic activities of electrocatalysts. Firstly, electrocatalysts with uniform dispersion and small particle sizes of active components can be obtained by plasma modification of the carrier or active component of electrocatalysts; secondly, plasma etching can be used to expose more interfacial defects on the surface of electrocatalysts and increase the catalytic active sites on the edge; thirdly, some heteroatoms (such as N, S, B, and P) can be doped by plasma to enhance the intrinsic defects, realizing the catalytic activity of electrocatalysts. Defects have been recognized as active sites with higher activity in electrocatalytic reactions (vacancies, marginal sites, and lattice defects). At present, how to precisely control and tailor the defects of the electrocatalysts, the surface modifications, and the atomic doping by plasma techniques, remains challenging.

## Figures and Tables

**Figure 1 nanomaterials-09-01436-f001:**
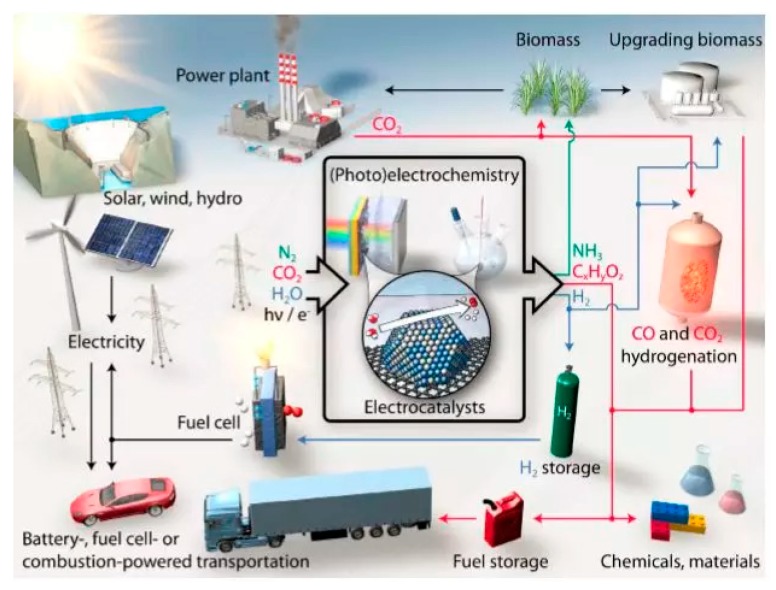
The roadmap of electrocatalytic reaction technology for sustainable energy use in the future [1]. (Reproduced with permission from [1]. American Association for the Advancement of Science, 2017).

**Figure 2 nanomaterials-09-01436-f002:**
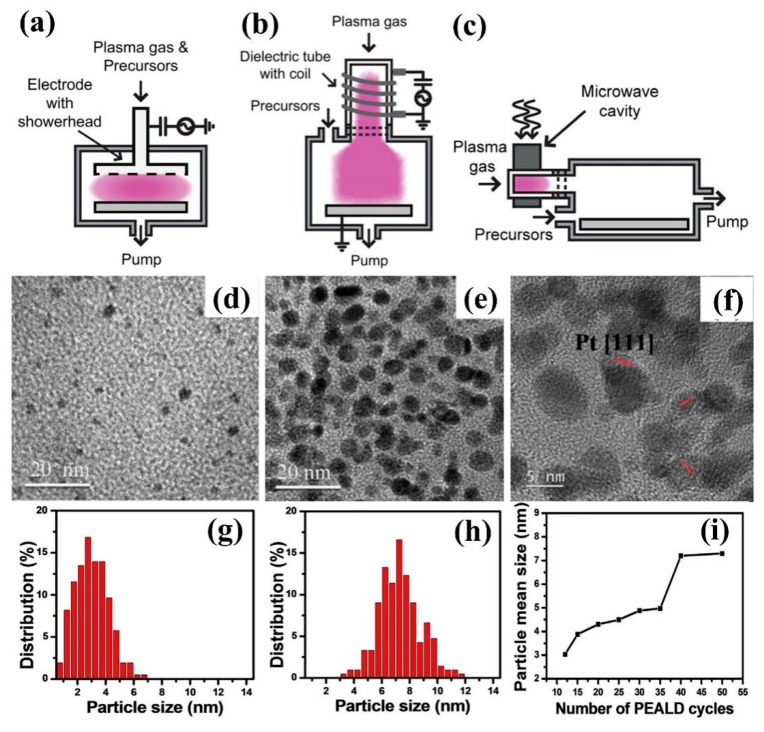
Plasma enhanced atomic layer deposition (PEALD) technology (**a**) direct plasma enhanced atomic layer deposition; (**b**) remote plasma enhanced atomic layer deposition; (**c**) radical-enhanced atomic layer deposition [13] (Reproduced with permission from [13], American Vacuum Society, 2007.) HRTEM images of samples with different PEALD deposition times: (**d**) 12 times, (**e**) 50 times, and (**f**) 40 times. Corresponding particle size distribution maps: (**g**) 12 times and (**h**) 50 times. (**i**) The graph of relationship between the average particle size distribution and the number of depositions [15]. (Reproduced with permission from [15]. Elsevier B.V., 2015).

**Figure 3 nanomaterials-09-01436-f003:**
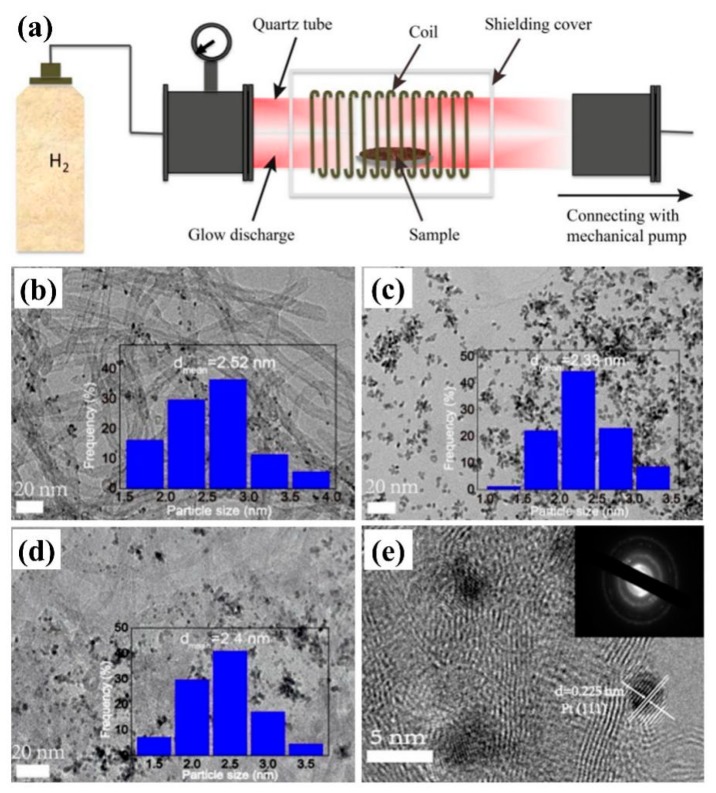
(**a**) The sketch of device for sample treatment by H_2_ plasma. TEM images and corresponding Pt particle size distribution images: (**b**) Pt/carbon nanotubes (CNTs), (**c**) Pt/RGO, and (**d**) Pt/GNT (graphene oxide (GO):CNTs = 1:2, mass ratio). (**e**) HRTEM and SAED images of Pt/GNT [20]. (Reproduced with permission from [20]. Elsevier B.V., 2018).

**Figure 4 nanomaterials-09-01436-f004:**
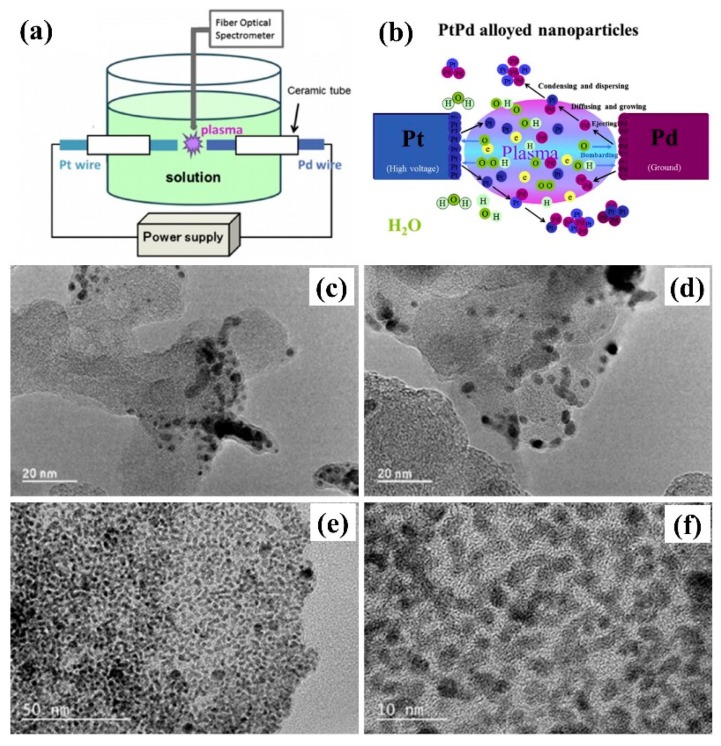
(**a**) The experimental device and (**b**) the schematic diagram of PtPb nanoparticles synthesized by solution plasma sputtering. Catalysts formed in different solutions: (**c**,**d**) PtPd/KB in aqueous solution; (**e**,**f**) PtPd/KB-2 in water-methanol mixture [30]. (Reproduced with permission from [30]. Elsevier B.V., 2017).

**Figure 5 nanomaterials-09-01436-f005:**
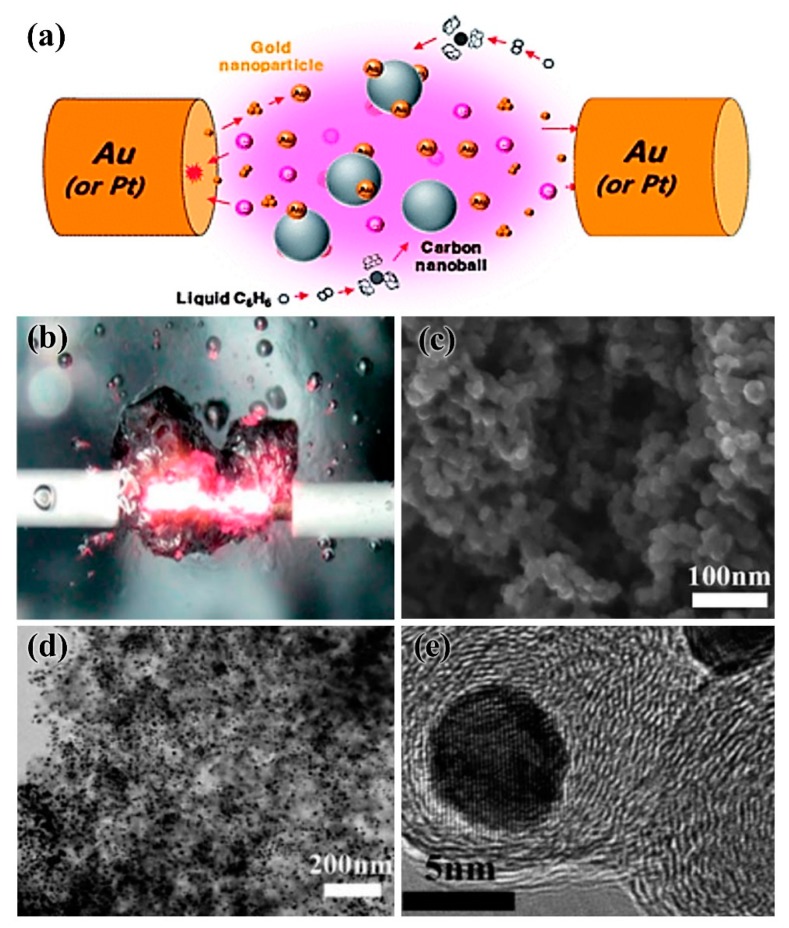
(**a**) The schematic diagram and (**b**) the physical diagram of carbon-loaded noble metal materials prepared by organic solution plasma. (**c**) Field emission scanning electron microscope (FESEM), (**d**) scanning transmission electron microscope (STEM), and (**e**) HRTEM diagrams of Au/CNBs catalysts [35]. (Reproduced with permission from [35]. Royal Society of Chemistry, 2013).

**Figure 6 nanomaterials-09-01436-f006:**
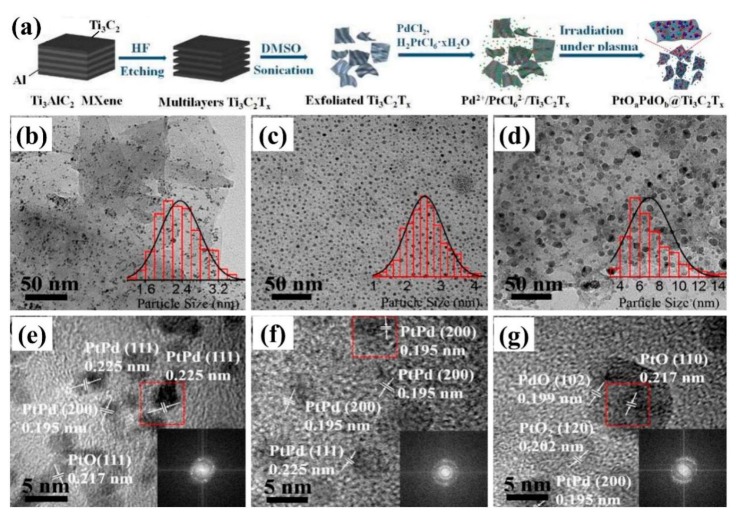
(**a**) The flow chart of PtO_a_PdO_b_@Ti_3_C_2_T_x_ catalysts prepared by liquid plasma reduction. TEM and HRTEM images with 200 W plasma for (**b**,**e**) 1 min, (**c**,**f**) 3 min, and (**d**,**g**) 5 min. (**e**–**g**) corresponding fast fourier transform (FFT) diffraction patterns [40]. (Reproduced with permission from [40]. American Chemical Society, 2018).

**Figure 7 nanomaterials-09-01436-f007:**
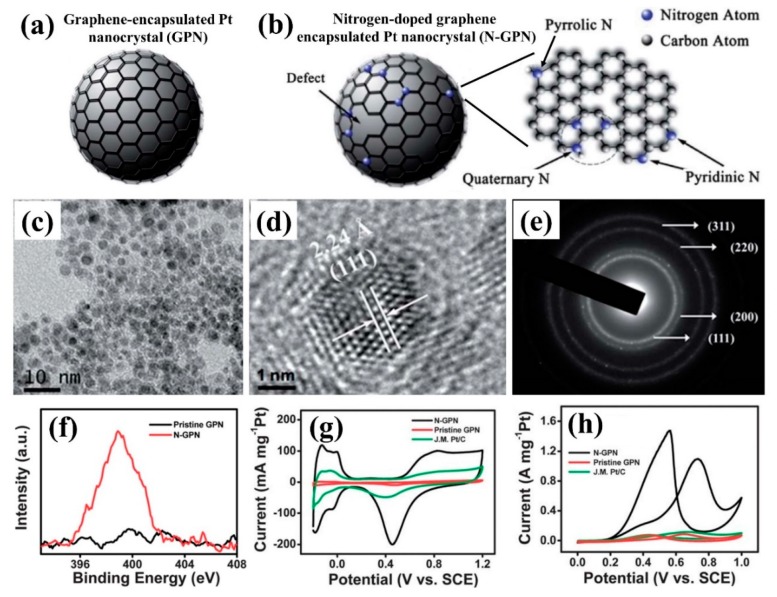
The schematic diagram of (**a**) graphene-supported Pt nanoparticles (GPN) and (**b**) N-doped GPN (N-GPN); The images of (**c**) TEM, (**d**) HRTEM, and (**e**) selected area electron diffraction (SAED) for GPN; (**f**) XPS image; (**g**) chemical vapor (CV)—0.5 M H_2_SO_4_, and (**h**) CV—0.5 M H_2_SO_4_ + 1.0 M CH_3_OH for GPN and N-GPN [45]. (Reproduced with permission from [45]. American Chemical Society, 2014).

**Figure 8 nanomaterials-09-01436-f008:**
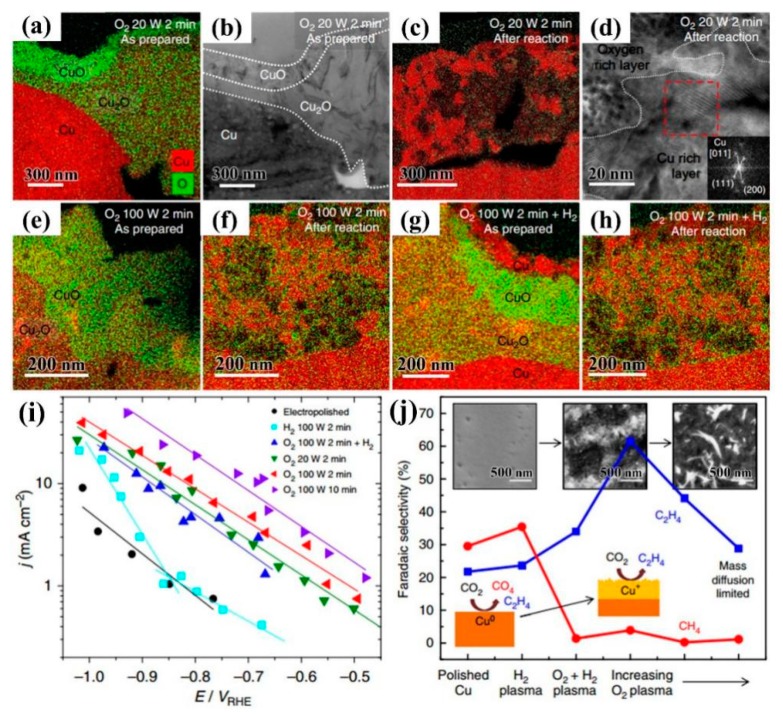
(**a**,**c**,**e**–**h**) EDS images and (**b**,**d**) HRTEM images of different plasma activated copper foil before and after reaction; (**i**) electrochemical activity during CO_2_RR process; (**j**) hydrocarbon selectivity of plasma treated Cu foil. The corresponding inset SEM images were after the reaction: H_2_ plasma treated metal Cu foil, O_2_ plasma treated with Cu foil at 20 W for 2 min, and O_2_ plasma treated at 100 W for 10 min, respectively [52]. (Reproduced with permission from [52]. Nature Publish Group, 2016).

**Figure 9 nanomaterials-09-01436-f009:**
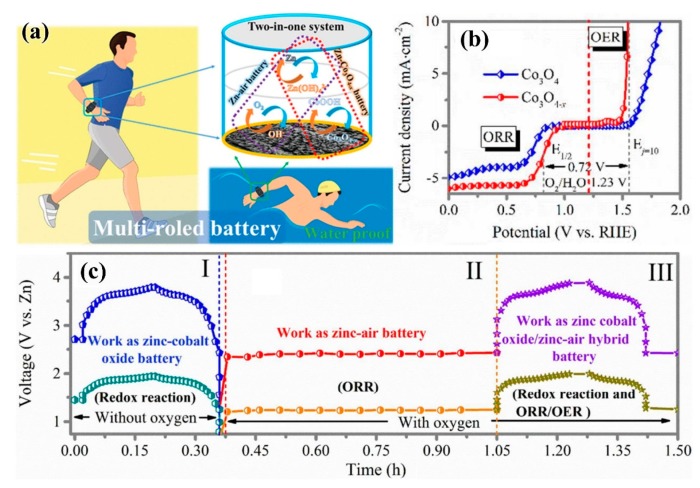
(**a**) The schematic diagram of Zn–Co_3_O_4−x_/Zn-air hybrid battery; (**b**) the overall polarization curve of Co_3_O_4_ nanorods and Co_3_O_4−x_; (**c**) the constant current charge/discharge curve of Zn–Co_3_O_4−x_ /Zn-air hybrid battery. (I) Work of the Zn–Co_3_O_4−x_ battery in an anaerobic environment; (II) Work of the zinc-air battery in an aerobic environment; (III) Work of the hybrid battery in an aerobic environment [62]. (Reproduced with permission from [62]. American Chemical Society, 2018).

**Figure 10 nanomaterials-09-01436-f010:**
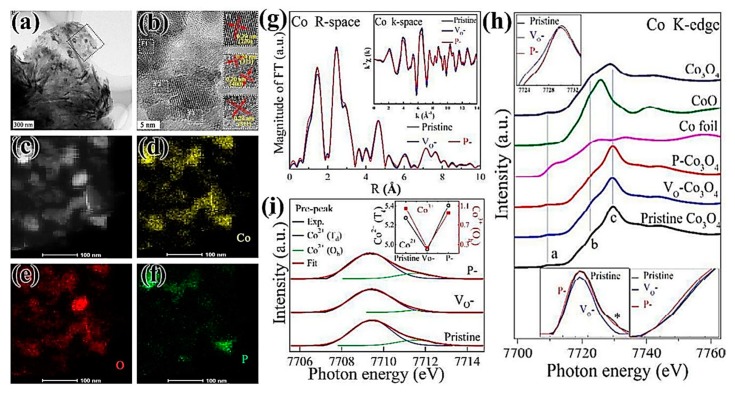
(**a**) TEM images, (**b**) HRTEM images, and (**c**–**f**) STEM-EDX elemental mapping images of P–Co_3_O_4_; (**g**) Co K-side extended x-ray absorption fine structure (EXAFS). The inset is the Fourier-transformed EXAFS oscillations; (**h**) Co K-edge XANES spectra of pristine V_0_–Co_3_O_4_ and P–Co_3_O_4_. Top inset magnifies the main peak region. Bottom insets magnify the pre-peak region; (**i**) deconvoluted pre-peak of Co K-edge XANES. Inset compares the amount of Co^2+^(Td) and Co^3+^(Oh) states of pristine, V_0_–Co_3_O_4_, and P–Co_3_O_4_ [67]. (Reproduced with permission from [67]. Royal Society of Chemistry, 2017).

**Figure 11 nanomaterials-09-01436-f011:**
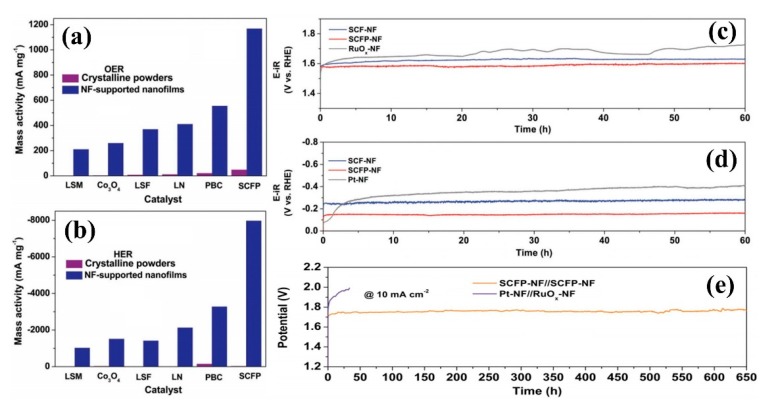
Comparison of the mass activity at an overpotential of 0.35 V between crystalline powders and NF-supported nanofilms for the oxygen evolution reaction (OER) (**a**) and hydrogen evolution reaction (HER) (**b**); the stability tests of SCFP-nickel foam (NF) and control samples for OER (**c**) and HER (**d**); (**e**) The stability tests for the water splitting of bifunctional SCFP-NF catalysts and Pt-NF coupled RuO-NF [73]. (Reproduced with permission from [73]. Wiley-VCH, 2018).

**Figure 12 nanomaterials-09-01436-f012:**
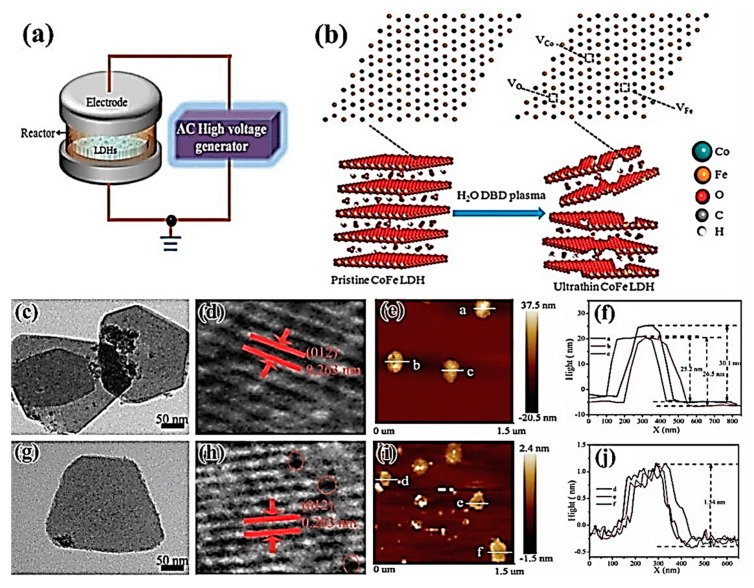
The schematic diagram of (**a**) a dielectric barrier discharge (DBD) reactor and (**b**) water DBD plasma-activated stripping of CoFe-LDHs nanosheets; TEM, HRTEM, and atomic force microscope (AFM) images and corresponding nanosheet thickness images of bulk CoFe-LDHs (**c**–**f**) and plasma-treated CoFe-LDHs–H2O samples (**g**–**j**) [74]. (Reproduced with permission from [74]. Royal Society of Chemistry, 2017).

**Figure 13 nanomaterials-09-01436-f013:**
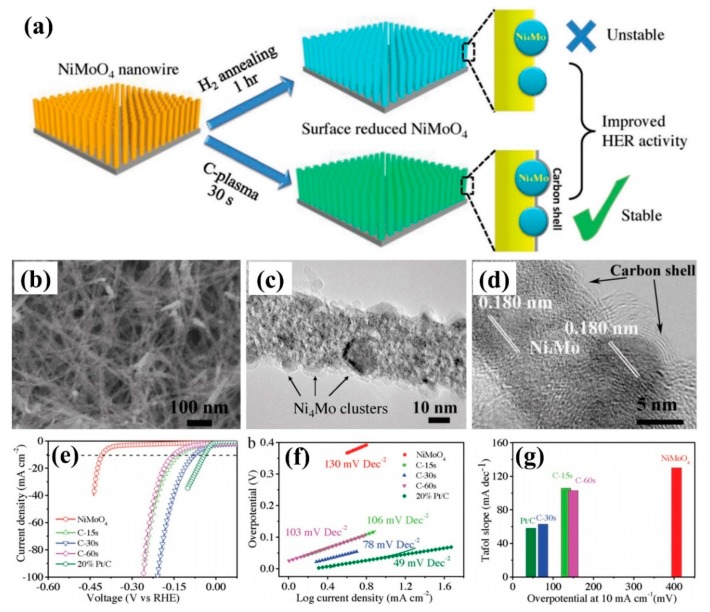
(**a**) The preparation of C-plasma treated and H_2_ annealed NiMoO_4_; (**b**) SEM and (**c**,**d**) TEM images of the sample C-30s; the electrochemical performance tests of the sample: (**e**) linear sweep voltammetry (LSV) curve, (**f**) Tafel slope, and (**g**) the relationship between overpotential and Tafel slope at current density of 10 mA/s [83]. (Reproduced with permission from [83]. Wiley-VCH, 2018).

**Figure 14 nanomaterials-09-01436-f014:**
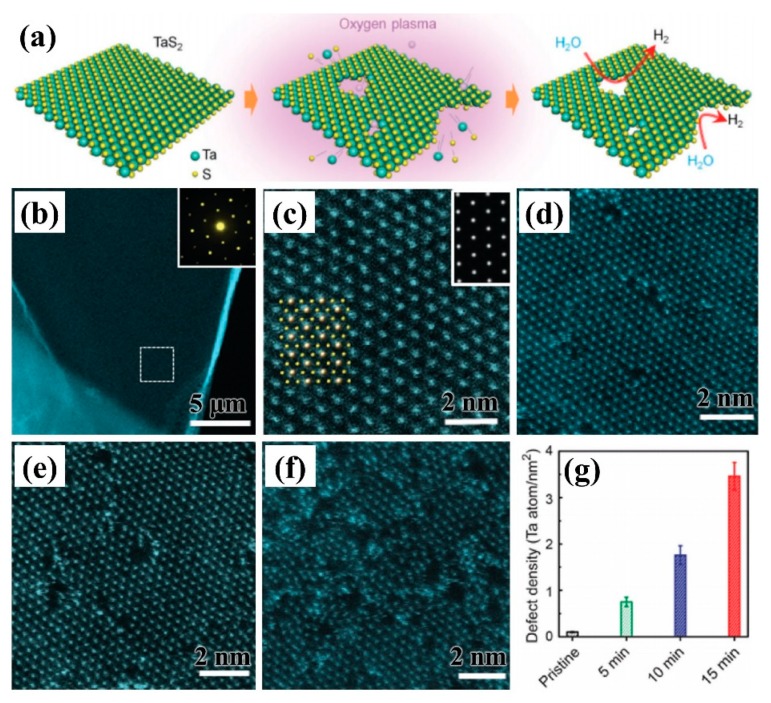
(**a**) Mechanism diagram of O_2_ plasma treated TaS_2_ nanosheets for HER; (**b**) TEM images and (**c**) HAADF-STEM images of TaS_2_; (**d**–**f**) HAADF-STEM images by plasma treatment for 5 min, 10 min, and 15 min; (**g**) the density of edge Ta atoms [85]. (Reproduced with permission from [85]. Wiley-VCH, 2016).

**Figure 15 nanomaterials-09-01436-f015:**
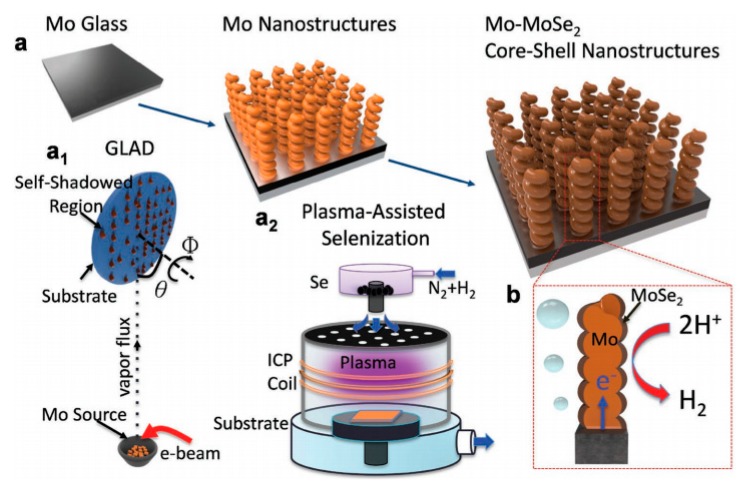
(**a**) Plasma-assisted selenization process: inset of (**a1**) MoSe_2_/Mo composite prepared by GLAD method; inset of (**a2**) plasma-assisted selenization treatment. (**b**) Schematic diagram of HER and charge transfer [99]. (Reproduced with permission from [99]. Wiley-VCH, 2016).

**Figure 16 nanomaterials-09-01436-f016:**
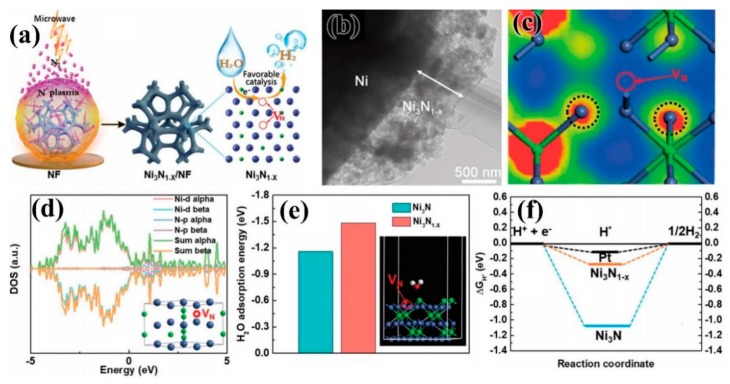
(**a**) Schematic diagram of preparation of self-supporting Ni_3_N_1−x_/NF electrode by microwave plasma-generation; (**b**) TEM images, (**c**) charge density distribution, and (**d**) the total electron density and partial electron density (TDOS and PDOS) of Ni_3_N_1−x_/NF; (**e**) adsorption energy of H_2_O molecules on the surface of Ni_3_N and Ni_3_N_1−x_; (**f**) free energy of H adsorption by various substances at equilibrium potential [102] (open access).

**Figure 17 nanomaterials-09-01436-f017:**
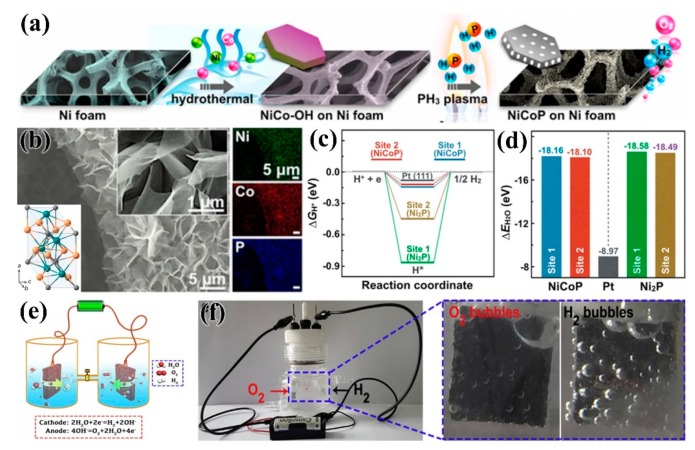
(**a**) Schematic diagram of the conversion of NiCo–OH nanosheets to NiCoP by PH_3_ plasma treatment; (**b**) SEM image, energy spectrum, and crystal structure of NiCoP; (**c**) free-energy diagram for H_2_ adsorption on the Ni_2_P, NiCoP (0001), and Pt (111) surfaces; (**d**) adsorption energy of water [108] (Reproduced with permission from [108]. American Chemical Society, 2016.) (**e**) Schematic diagram of the bifunctional electrocatalysts OER and HER; (**f**) photograph of the water splitting in the alkaline solution at 1.5 V [113] (Reproduced with permission from [113]. Elsevier B.V., 2018).

**Figure 18 nanomaterials-09-01436-f018:**
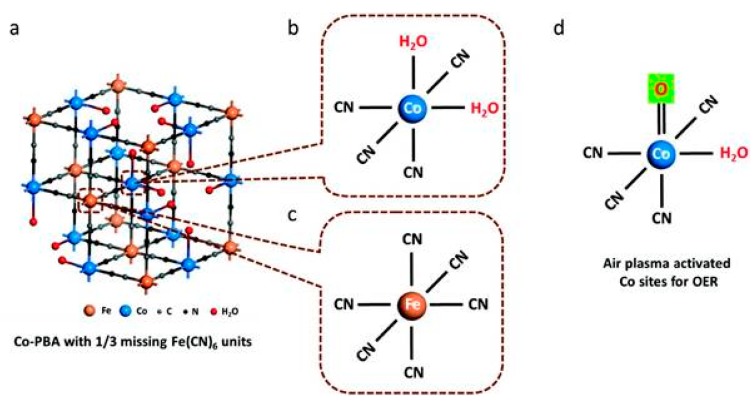
(**a**) The structure of Co–PBA with the composition of Co_3_(Fe(CN)_6_)_2_; the coordination structure of (**b**) Co and (**c**) Fe sites. Each Co center has two open sites, which are occupied by coordinated water molecules, while the Fe center is completely coordinated by six CN groups; (**d**) schematic diagram of metal sites in Prussian blue’s structure modified by air plasma [117]. (Reproduced with permission from [117]. Wiley-VCH, 2018).

**Figure 19 nanomaterials-09-01436-f019:**
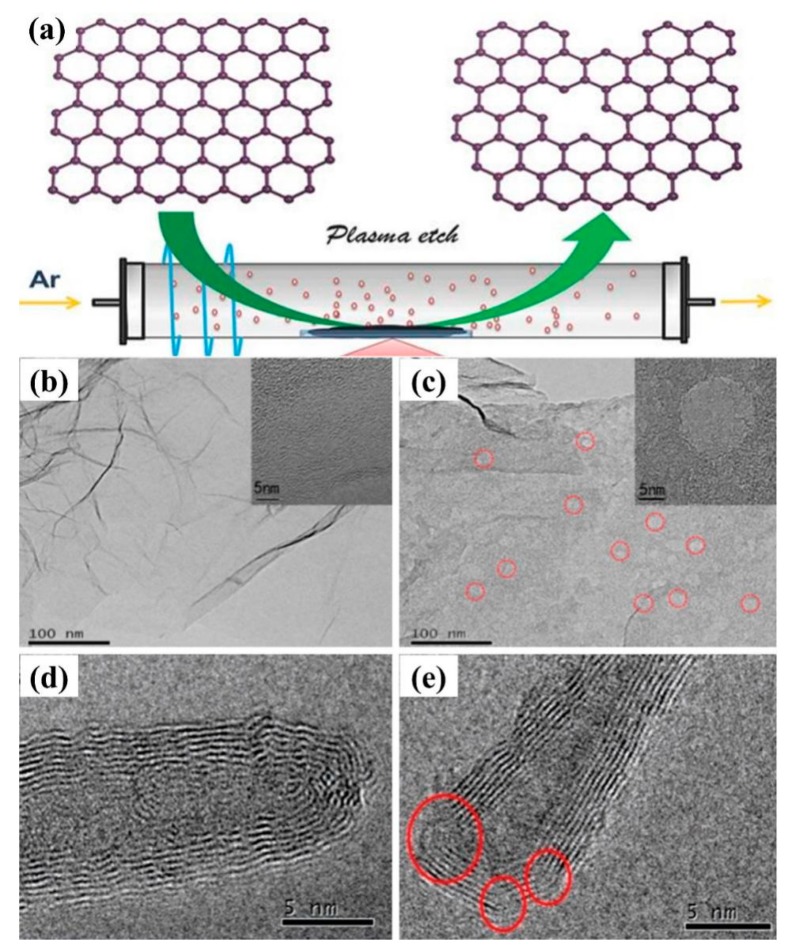
(**a**) Schematic diagram of the preparation of Ar plasma etching graphene and carbon nanotubes surface by Ar plasma; TEM images of (**b**) graphene and (**c**) Ar plasma treated graphene; TEM images of (**d**) carbon nanotubes and (**e**) Ar plasma treated carbon nanotubes [124]. (Reproduced with permission from [124]. Royal Society of Chemistry, 2016).

**Figure 20 nanomaterials-09-01436-f020:**
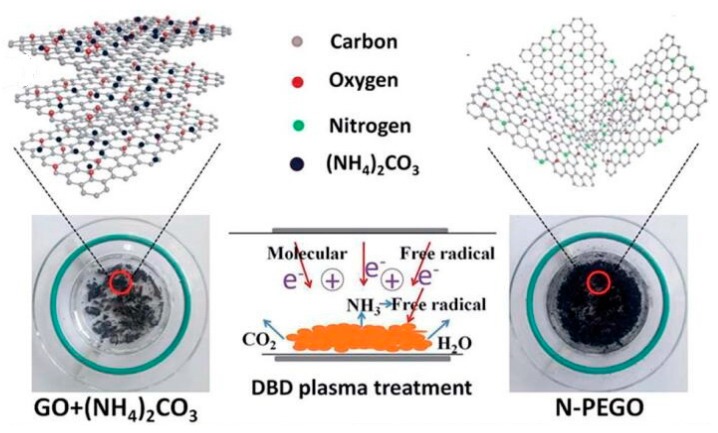
Schematic diagram of N-doped graphene oxide prepared by low temperature plasma technology [127]. (Reproduced with permission from [127]. Royal Society of Chemistry, 2018).

**Figure 21 nanomaterials-09-01436-f021:**
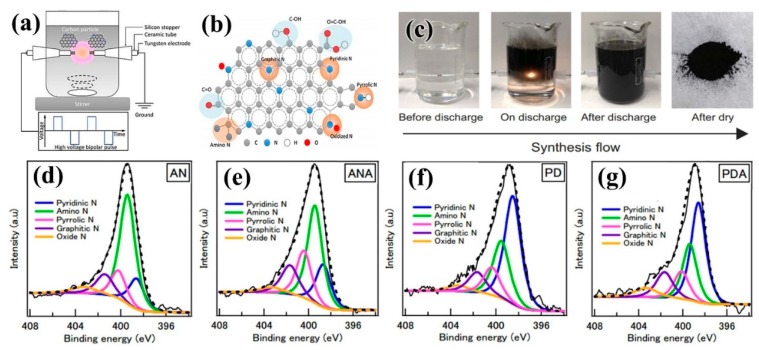
(**a**) Schematic diagram of preparation of N-doped carbon nanoparticles (NCNP) by solution plasma method; (**b**) species of doped single atoms [135] (copyright @ 2017 Royal Society of Chemistry.) (**c**) Process for preparing nano-carbon materials by solution plasma method [139] (open access.) (**d**–**g**) Distribution of nitrogen elements of NCNP prepared from acrylonitrile (AN), acrylonitrile + ANA, pyridine (PD), and pyridine + hydrazine (PDA) as precursors [135]. (Reproduced with permission from [135]. Royal Society of Chemistry, 2017).

**Figure 22 nanomaterials-09-01436-f022:**
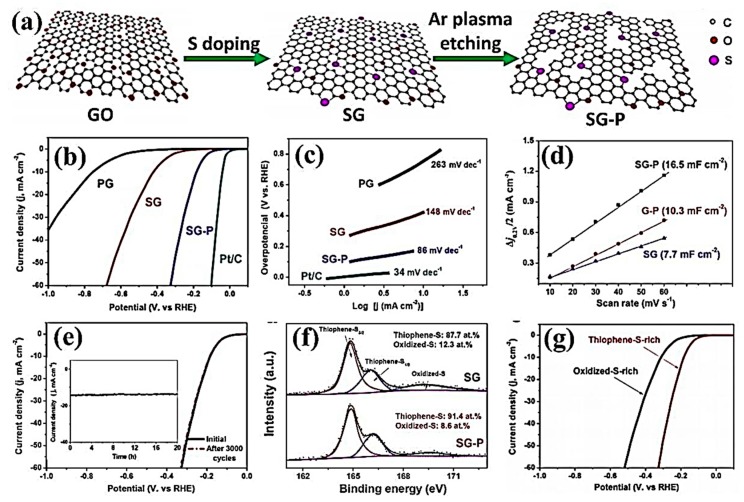
(**a**) Schematic diagram of the treatment of S-doped graphene BY Ar plasma. (**b**) Polarization potential and (**c**) Tafel slope at a scan speed of 5 mV/s in a 0.5 M H2SO4 solution of PG, SG, SG–P, and commercial 20% Pt/C. (**d**) Current densities of SG, G–P, and SG-P. (**e**) Cycle performance of SG–P. (**f**) PG, SG, SG–P, and commercial 20% Pt/C of S’s 2p for SG and SG–P. (**g**) Different polarization curves for SG–P enriched in thiophene sulfur and oxidized sulfur [143]. (Reproduced with permission from [143]. Elsevier B.V., 2017).

**Figure 23 nanomaterials-09-01436-f023:**
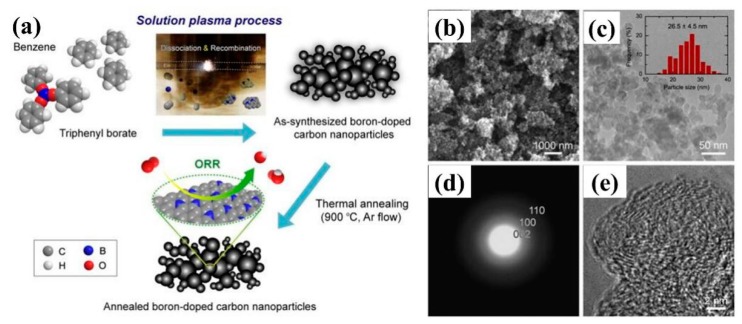
(**a**) Schematic diagram of the preparation of B-doped nanoparticles (BCNP) by solution plasma. (**b**) SEM image, (**c**) TEM images, and particle size distribution image; (**d**) SAED diffraction pattern; and (**e**) HRTEM images of BCNP [145]. (Reproduced with permission from [145]. Elsevier B.V., 2015).

**Table 1 nanomaterials-09-01436-t001:** Preparation of electrocatalysts by plasma technology.

Reaction Type	Samples	Methods	Electrochemical Performance	Ref.
MOR	Pt/TiO_2_	PEALD	the MOR current density drops to a small value after 1500 s with NCALD < 30.	[15]
	Pt/C	CAPD	The calculated ECSAs of 75.4 m^2^/g	[16]
	Pt/GNT	H_2_ plasma	The current density of 97.9 mA/mg and mass activity of 691.1 mA/mg Pt	[20]
	Pt/CNTs-HP	H_2_ plasma	The current density of 15.8 mA/mg	[21]
	Au@Pt	Ar plasma	Mass activity up to 48 ± 3 m^2^/g	[23]
	Pt-Ag	SPS	The calculated ECSAs of 28.15 m^2^/g	[27]
	Pt_40_Pd_60_	SPS	The calculated ECSAs of 28.15 m^2^/g	[28]
	Pt_69_Pd_31_	SPS	The catalytic activity of 6.81 mA/cm^2^	[29]
	PtPd/KB-2	SPS	The electrocatalytic activity was more than 4 times of that of commercial Pt/C	[30]
	Pt/CoPt/MWCNS	SPS	Mass activities of 1719 mA/mg Pt	[32]
	Pt/C/TiO_2_-2	SPS	Mass activities of 315.2 mA/mg Pt	[33]
	Pt-Ru/OCNT	O_2_ plasma	The onset potential of 0.3 V	[44]
	Pt/ZnO/KB	SPS	Catalytic activity of 964 mA/mgPt	[48]
OER	Pt-Ir/TiC	Ar plasma	The current density of 2.5 mA/cm^2^ at 1 V	[22]
	PtO_a_PdO_b_@Ti_3_C_2_T_x_	SPS	Activation potential of 1.5 V in 0.1 M KOH	[40]
	Ag/1400-15	Spark plasma sintering	The value of 61 mV	[41]
	CoNPs@ C	MPECVD	Overpotential of 270 mV	[50]
	CoO–N/C	Cold plasma deposition	The oxygen evolution potential of 378 mV	[55]
	Co_3_O_4_ nanosheets	Ar plasma	The current density of 44.44 mA/cm^2^ at 1.6 V	[60]
	V_o_-COOH	Ar plasma	The low overpotential of 262 mV at 10 mA/cm^2^	[61]
	Co_3_O_4−x_	Ar plasma	The overpotential of 330 mV and Tafel slope of 58 mV/dec	[62]
	N-Co_3_O_4_	N_2_ plasma	The required potential of 1.54 V to reach the current density of 10 mA cm^−2^	[64]
	P-Co_3_O_4_	Ar plasma	The overpotential of 280 mV and Tafel slope of 51.6 mV/dec	[67]
	CoO_x_-ZIF	O_2_ plasma	The required potential of 1.548 V to reach the current density of 10 mA cm^−2^	[69]
	SnCoFe-Ar	Ar plasma	The overpotential of 300 mV and Tafel slope of 42.3 mV/dec	[71]
	SrTi_0.5_Fe_0.5_O_3−y_	HT-PVD using O_2_ plasma	The onset potential of 1.40 at 100 μA	[72]
	SCFP-NF	High-energy argon plasma sputtering	The ultra high mass activity of 1000 mA/mg	[73]
	CoFe-LDHs-H_2_O	H_2_O plasma	The overpotential of only 232 mV	[74]
	CoFe-LDHs-Ar	Ar plasma	The overpotential of only 232 mV at a current density of 10 mA/cm^2^	[77]
	CoFe-LDHs-N_2_	N_2_ plasma	The overpotential of only 233 mV at a current density of 10 mA/cm^2^	[78]
	FeO_x_/C	pulsed arc plasma deposition	The discharge specific capacity of 500 mAh/g under 100 mA/g	[80]
	Co_9_S_8_/G	NH_3_ plasma	The Tafel slope of 82.7 mV/dec	[91]
	Cu_2_S/CF	active iodine DBD plasma	The overpotential of 290 mV at a current density of 10 mA/cm^2^	[93]
	CoS/Ni_3_S_2_-FeS/PNFF	Air plasma	The overpotential of 136 mV at a current density of 10 mA/cm^2^	[96]
	hNi_3_N	N_2_ plasma	The overpotential of 325 mV at a current density of 10 mA/cm^2^	[101]
	NiCoP	PH_3_ Plasma	The overpotential of 280 mV at a current density of 10 mA/cm^2^	[108]
	NiFePi/P	PECVD	The overpotential of 230 mV at a current density of 10 mA/cm^2^	[109]
	O_3_-V_10_-Ni_2_P	O_2_ plasam	The Tafel slope of 43.5 mV/dec	[110]
	PA-CoPO	H_2_ plasma	The overpotential of 240 mV and Tafel slope of 53 mV/dec	[113]
	Co-PBA-plasma 2 h	Air plasma	The overpotential of 330 mV at a current density of 100 mA/cm^2^	[117]
	NGF-CFP	N_2_ plasma	The current density of 10 mA/cm^2^ at 2.16 V	[128]
SO_2_OR	Pt_3_Pd_2_ and PtPd_4_	Plasma sputtering	The lower onset potential of 0.587 ± 0.004 V	[18]
HOR	Pt/C	SPS	The ECSAs value of 266 cm^2^/mg	[38]
AMXOR	Ti_4_O_7_	Plasma deposition	A quick oxidation of 0.1 mM AMX	[79]
ORR	Pt/C	CAPD	The half-wave potential of 0.87 V	[16]
	Pt-Ir/TiC	Ar plasma	Nyquist plots of 0.6 V	[22]
	Au@Pt	Ar plasma	The peak potential of 0.75 V	[23]
	Pt/XC72	SPS	The onset potential of 1.04 V	[31]
	PdAu/KB	SPS	The reduction peak disappeared after about 700 cycles	[34]
	AgNW/C	SPS	The high electron density of 13.7 × 10 − 22 m^−3^	[37]
	Pt/rGO-N	SPS	The half-wave potential of 0.87 V	[39]
	Ag/1400-15	spark plasma sintering	The electron transfer number of 3.9	[41]
	PtNW/GDL	N_2_ + H_2_ plasma	The power density of 64 mW/cm^2^	[47]
	CoO–N/C	Cold plasma deposition	A 2-electron process producing H_2_O_2_	[55]
	Ag@Co_3_O_4_	SPS	The half-wave potential of 0.799 V	[56]
	15Co_3_O_4_/N-AP/800	microwave-induced plasma	The Tafel slope of 42 mV/dec	[57]
	Co-La-Pt	DC arc discharge plasma	The specific capacity of 3250.2 mAh/g and energy density of 8574.2 Wh/kg at 0.025 mA/cm^2^	[58]
	Co_3_O_4−x_	Ar plasma	The half-wave potential of 0.84 V	[62]
	FeO_x_/C	Pulsed arc plasma deposition	The discharge specific capacity of 500 mAh/g under 100 mA/g	[80]
	MnO_x_@C-D	Air plasma	The electron transfer number of 3.81	[81]
	A-MnO_2_	Ar plasma	The power density of 159 mW/cm^2^ at a current density of 157 mA/cm^2^	[82]
	Cu_3_N_200_/C	PEALD	The half-wave potential of 0.684 V	[103]
	Fe-N-CNP-CN	SPS	The onset potential of −0.10 V	[104]
	Fe–N/C	Air plasma	The onset potential of 0.88 V at a loading of 0.60 mg/cm^2^	[105]
	P-Graphene	Ar plasma	The onset potential of 0.912 V and half-wave potential of 0.737 V	[124]
	P-CNTs	Ar plasma	The onset potential of 0.83 V	[124]
	P-CC	Ar Plasma	The exchange current density of 2.57 × 10^−9^ A/cm^2^	[125]
	hCNW-60	PECVD	The onset potential of 830 mV in 0.1 KOH	[126]
	N-PEGO	DBD plasma	The onset potential of 0.89 V	[127]
	NCNTs	MPCVD	The onset potential of 0.87 V and the electron transfer number of 4.1	[130]
	VA-NCNTs	N_2_ plasma	The ORR peak at the potentials of about −0.3 V	[131]
	NCNPs	SPS	The onset potential of −0.17 V	[137]
	NCNP-3	SPS	The onset potential of −0.143 V	[138]
	BZ90 + DO10	SPS	The samples were held at 0.5 V with a rotation speed of 1500 rpm for 45,000 s in an O_2_-saturated 0.1 M KOH solution	[140]
	O-rGO	O_2_ Plasma	The current density of 10 µA/cm^2^	[141]
	BCNP	SPS	The current densities of 3.15 mA/cm^2^ at −0.60 V	[145]
	FCNP-4	SPS	The onset potential of 0.22 V and limiting current density of 2.76 mA/cm^2^ at 0.6 V	[147]
	BCN nanocarbon	SPS	15.1% current decrease after 20000 s	[149]
CO_2_RR	Au island	O_2_ plasma	The faradaic efficiency over 95%	[24]
	CNT/Cu	O_2_ plasma	Carbon monoxide yields of 178 mmol cm^2^ mA^−1^ h^−1^	[51]
	Cu foil	O_2_ plasma	The ethylene selectivity of 60%	[52]
	Cu nanocube	O_2_ plasma	The ethylene selectivity of 60%	[53]
	ZnO	H_2_ plasma	The current density of −16.1 mA/cm^2^ and faradaic efficiency of 83%	[84]
	PEI-NCNT/GC	NH_3_ plasma	The current density of 2.2 mA/cm^2^	[132]
HER	Ni–Fe–C	CH_4_+H_2_ Plasma carburizing	The activation potential of 57 mV at a current density of 10 mA/cm^2^	[49]
	CoNPs@ C	MPECVD	The overpotential of 153 mV at a current density of 10 mA/cm^2^	[50]
	P-Co_3_O_4_	Ar plasma	The overpotential of 120 mV and Tafel slope of 52 mV/dec	[67]
	SCFP-NF	high-energy argon plasma sputtering	The onset potential of −0.01 V and Tafel slope of 94 mV/dec	[73]
	C-30s	C plasma	The Tafel slope of 44 mV/dec	[83]
	TaS_2_-15 min	O_2_ plasma	The onset potential of 310 mV	[85]
	O_2_-MoS_2_	O_2_ plasma	The current density of 16.3 mA/cm^2^ at −350 mV	[86]
	MoS_2_	O2 plasma	The overpotential of 131 mV at a current density of 10 mA/cm^2^	[87]
	H_3_Mo_12_O_40_P/MoS_2_	O_2_ plasma	The Tafel slope of 44 mV/dec	[88]
	MoS_2_-15 min	H_2_ plasma	The overpotential of 240 mV at a current density of 10 mA/cm^2^	[89]
	MoS_1.7_	H_2_ plasma	The overpotential of 143 mV at a current density of 10 mA/cm^2^	[90]
	Co_3_S_4_ PNSvac	Ar plasma	The mass activity of 1056.6 A/g at an overpotential of 200 mV	[92]
	WS_2_	PEALD	The overpotential of 90 mV at a current density of 100 mA/cm^2^	[94]
	WS_2_	SF_6_/C_4_F_8_ plasma-etched	The overpotential of 100 mV and Tafel slope of 50 mV/dec	[95]
	CoS/Ni_3_S_2_-FeS/PNFF	Air plasma	The overpotential of 75 mV at a current density of 10 mA/cm^2^	[96]
	N-MoSe_2_/VG	MPECVD	The onset potential of 45 mV and overpotential of 98 mV at 10 mA/cm^2^	[98]
	MoSe_2_/Mo	N_2_/H_2_ plasma	The Tafel slope of 34.7 mV/dec	[99]
	Ni_3_N_1−x_/NF	Microwave plasma	The overpotential of 55 mV and Tafel slope of 54 mV/dec at a current density of 10 mA/cm^2^	[102]
	NiCoP	PH_3_ plasma	The overpotential of 32 mV at a current density of −10 mA/cm^2^	[108]
	O_3_-V_10_-Ni_2_P	O_2_ plasma	The overpotential of 108 mV and Tafel slope of 72.3 mV/dec	[110]
	Ni-FeP/TiN/CC	Plasma-implanted method	The overpotential of 75 mV at a current density of 10 mA/cm^2^	[111]
	CoP_x_	PEALD	The exchange current density of −8.9 × 105 A/cm^2^	[112]
	PA-CoPO	H_2_ plasma	The overpotential of 50 mV at a current density of 10 mA/cm^2^	[113]
	WC nanowalls	DC-PACVD	The Tafel slope of 67 mV/dec	[116]
	Co-PBA-plasma 2 h	Air plasma	The overpotential of 77 mV at a current density of 20 mA/cm^2^	[117]
	SG-P	Ar plasma	The overpotential of 178 mV at a current density of 10 mA/cm^2^	[143]
	3DSG-Ar	Ar plasma	The Tafel slope of 64 mV/dec	[144]
	P-NSG	Ar plasma	The onset potential of 58 mV	[148]

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
