# Peer review of "A Review on the Promising Plasma-Assisted Preparation of Electrocatalysts"

_nanomaterials, 2019, doi:10.3390/nano9101436_

Round 1

Reviewer 1 Report
Great paper , references up to date

Author Response

Thank you for your approval. We appreciate it.

Reviewer 2 Report

I attach file with comments

Reviewer 3 Report

The authors have done a nice job in reviewing plasma effects in electro-catalyst nanomaterials. The comprehensive summary Table should prove very useful to the readers of the paper. Also, a good fraction of the references have been published during the past few years (2017, 2018, 2019) and this provides a clear picture of advances in the field. I recommend publication in Nanomaterials after the authors make a revision to correct language and non-technical mistakes in the manuscript.

Line 60, replace “generte” by “generate” Line 187, should CNB be CNS for carbon nanospheres? Lines 188-190, the following text is repeated verbatim: “At the same time, metal nanoparticles (such as Au, Pt) produced by metal electrode sputtering are deposited on carbon nanospheres to obtain highly active electrocatalysts loaded with precious metal nanoparticles.” Line 193, particle diameter 20-30 nm and pore diameter 13-16 nm implies that the particles is one or two pores. Please check. Line 216 replace “electrolyze water” by “water electrolysis” Line 340 replace “which may resulted” by “which may have resulted” Line 365, please define ZIF Line 372, please define MOFs Lines 388-390 are a verbatim repetition of lines 371-373. Line 391 replace “tyoe” by “type Line 402 replace “selectively preferentially” by “preferentially” Line 403 replace “had” by “has” Lines 408-411, text is verbatim repetition of text in lines 404-407 Line 429 replace “2.1 times” by ”2.1 times higher” Line 481 remove the word “potential” Line 493 replace “100 mA/g” by “100 Ma/cm2” Line 506 remove the word “layer” Line 516, what is the meaning of negative current density? Line 563, please define PNSvac Line 575, remove the word “potential” Line 583, define PNFF Line 599, define 2H-1T Line 638, what does “carbon-enca” mean? Line 665, define DOS Line 695, replace “H-superscript 2” by “H-subscript 2” Line 865, remove “overpotential of 178 mV Line 867, should “effects” be “defects”? Table 26 lines from the top, please give units of 1.4 Page 35 under Author Contributions replace “writed” by “wrote” Page 35, replace “Referces” by “References”

Reviewer 4 Report

This review is well organized for publication in Nanomaterials.

Author Response

(The authors gave the same response as above.)
